# LOCATE 3D: Real-World Object Localization via Self-Supervised Learning in 3D

Paul McVay [* 1]   Sergio Arnaud [* 1]   Ada Martin [1]   Arjun Majumdar [1]   Krishna Murthy Jatavallabhula [1]
Phillip Thomas [1]   Ruslan Partsey [1]   Daniel Dugas [1]   Abha Gejji [1]   Alexander Sax [1]   Vincent-Pierre Berges [1]
Mikael Henaff [1]   Ayush Jain [1 2]   Ang Cao [1 3]   Ishita Prasad [1]   Mrinal Kalakrishnan [1]   Michael Rabbat [1]
Nicolas Ballas [1]   Mido Assran [1]   Oleksandr Maksymets [1]   Aravind Rajeswaran [1]   Franziska Meier [1]

## Abstract

We present LOCATE 3D, a model for localizing objects in 3D scenes from referring expressions like "the small coffee table between the sofa and the lamp." LOCATE 3D sets a new state-of-the-art on standard referential grounding benchmarks and showcases robust generalization capabilities. Notably, LOCATE 3D operates directly on sensor observation streams (posed RGB-D frames), enabling real-world deployment on robots and AR devices. Key to our approach is 3D-JEPA, a novel self-supervised learning (SSL) algorithm applicable to sensor point clouds. It takes as input a 3D pointcloud featurized using 2D foundation models (CLIP, DINO). Subsequently, masked prediction in latent space is employed as a pretext task to aid the self-supervised learning of contextualized pointcloud features. Once trained, the 3D-JEPA encoder is finetuned alongside a language-conditioned decoder to jointly predict 3D masks and bounding boxes. Additionally, we introduce LOCATE 3D DATASET, a new dataset for 3D referential grounding, spanning multiple capture setups with over 130K annotations. This enables a systematic study of generalization capabilities as well as a stronger model. Code, models and dataset can be found at the project website: locate3d.atmeta.com

## 1. Introduction

For AI systems to effectively assist us in the physical world, such as through robots or smart glasses, an understanding of the 3D world grounded in human natural language is essential. Towards this goal, we study the task of 3D localization via referring expressions (Chen et al., 2020; Achlioptas et al., 2020), or simply 3D-REFEXP. It requires localizing an object (in 3D) from a textual *expression* that may include a combination of attributes (e.g., *"red chair"*) and/or spatial relationships (e.g., *"the small coffee table between the sofa and the lamp"*). In this work, we develop LOCATE 3D (see Figure 1), a state-of-the-art model for 3D-REFEXP. It builds on two key components: (1) 3D-JEPA, a novel SSL algorithm to learn contextualized scene representations, and (2) a novel language-conditioned 3D localization decoder.

The task of 3D-REFEXP is challenging to date. At one end of the spectrum are methods that train specialized models for this task (Jain et al., 2021; Chen et al., 2022) on small benchmark datasets. They often require human annotation at inference time in the form of detailed 3D meshes or object instance segmentation, making them difficult to deploy on real-world devices. At the other end are methods that try to leverage 2D VLMs for 3D tasks (Xu et al., 2024; Zhu et al., 2023). While these methods can encode rich linguistic structures by leveraging LLMs, they employ a simplistic and hand-crafted representation of the 3D world.

LOCATE 3D operates in three phases. In the first **pre-processing** phase, we leverage the underlying sensor observation stream to lift features from 2D foundation models (Radford et al., 2021; Oquab et al., 2023) into 3D point clouds (Jatavallabhula et al., 2023). Subsequently, we use a transformer encoder, pre-trained with our 3D-JEPA SSL algorithm, to transform the "lifted" foundation features into **contextualized features** that can provide better scene-level understanding. Finally, we use a language conditioned **3D decoder** (Cheng et al., 2021; Kamath et al., 2021) to localize the object of interest. Notably, LOCATE 3D operates directly on sensor observation streams without requiring manual post-processing (e.g., 3D mesh refinement or ground-truth instance segmentations), making it readily deployable on robots and AR devices.

We outline **our contributions** in this work below.

1. **3D-JEPA** – A novel SSL method applicable to 3D point

---

[*]Equal contribution  [1]FAIR at Meta [2]Carnegie Mellon University [3]University of Michigan, Ann Arbor. Correspondence to: Sergio Arnaud <sergioarnaud@meta.com>, Paul McVay <pmcvay@meta.com>.

*Proceedings of the 42nd International Conference on Machine Learning*, Vancouver, Canada. PMLR 267, 2025. Copyright 2025 by the author(s).

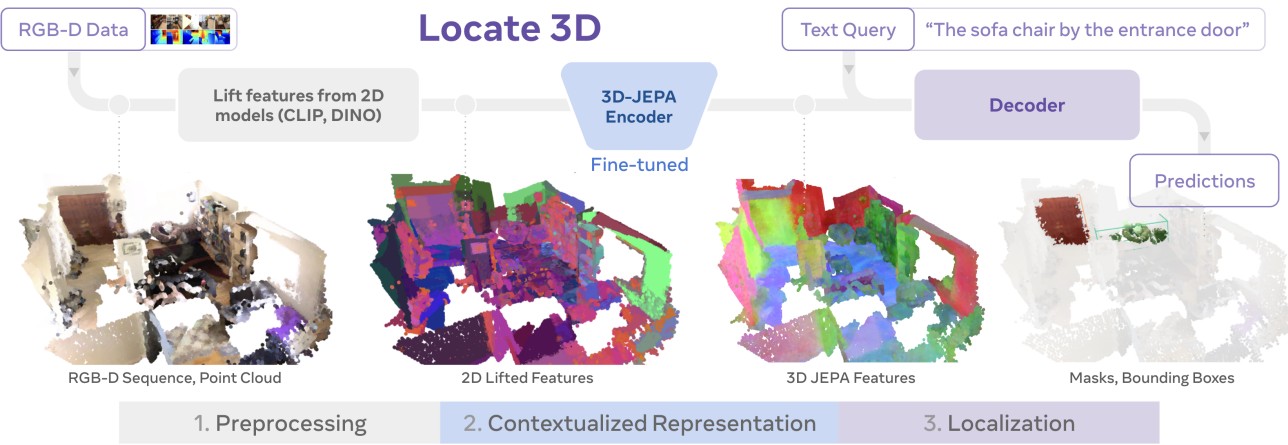

Figure 1: **Overall Architecture of LOCATE 3D**, which operates in three phases. In **Phase 1: Preprocessing**, we construct a point cloud with "lifted" features from 2D foundation models, which provide local information. In **Phase 2: Contextualized Representations**, these lifted features are passed through the pre-trained 3D-JEPA encoder, which provides a contextualized representation for the whole scene. Finally, in **Phase 3: 3D Localization**, a 3D decoder head uses the text query and 3D-JEPA features to localize the referred object.

clouds that can learn contextualized representations of 3D scenes. It takes as inputs 3D point clouds with features lifted from 2D foundation models. The SSL pretext task involves predicting the latent embeddings of randomly masked regions in the featurized point cloud (Assran et al., 2023). We show that the resulting 3D-JEPA features are contextualized for the scene, while the features lifted from 2D foundation models only provide local understanding. Conceptually, this is analogous to the difference between contextualized token embeddings (Devlin et al., 2019) and word embeddings (Mikolov et al., 2013) in NLP. 3D-JEPA pre-training provides a significant performance gain to our LOCATE 3D model, for both in-domain (**59.8%** $\rightarrow$ **61.7%**) and out-of-domain (**51.5%** $\rightarrow$ **56.7%**) evaluations.

2. **LOCATE 3D** - a model for 3D RefExp (see Figure 1) that achieves SoTA benchmark results and strong out-of-domain generalization. LOCATE 3D leverages our 3D-JEPA and fine-tunes it for the 3D RefExp task using a language-conditioned 3D decoder. The decoder performs interleaving cross-attention between the 3D features and text queries, and employs a joint mask and bounding box prediction strategy. On standard 3D RefExp benchmarks (ScanRefer, SR3D, NR3D), LOCATE 3D achieves SoTA results compared to prior work (**58.5%** $\rightarrow$ **61.7%**). Crucially, LOCATE 3D achieves these impressive results with fewer assumptions compared to prior models. LOCATE 3D does not require ground-truth region proposals, meshes, or surface normals at inference time, making it suitable for real-world deployment. When comparing to prior work under similar assumptions (Jain et al., 2021), LOCATE 3D presents a significant advancement (**40.7%** $\rightarrow$ **61.7%**). It further exhibits strong general-

ization capabilities on held-out scenes and annotations in ScanNet++. Finally, it also successfully localizes objects in a multi-room test environment enabling robotic mobile manipulation in an unseen environment.

3. **LOCATE 3D DATASET (L3DD)** - A new dataset for 3D RefExp which spans ScanNet (Dai et al., 2017), ScanNet++ (Yeshwanth et al., 2023), and ARKitScenes (Dehghan et al., 2021). L3DD covers 1,346 scenes and over 130K language annotations. It allows us to study the robustness of LOCATE 3D to a variety of capture setups and independent samplings of indoor environments. It also serves as a source of training data for RefExp models; while LOCATE 3D already achieves SoTA performance using only standard benchmark training datasets (**61.7%**), L3DD training data further improves performance of our approach on these benchmarks (**61.7%** $\rightarrow$ **63.7%**). We call this model LOCATE 3D+.

## 2. LOCATE 3D: Model Overview and Training

The overall architecture of LOCATE 3D is presented in Figure 1. LOCATE 3D is designed to operate on RGB-D sensor observations of static environments (e.g., homes in which objects remain stationary over short intervals). LOCATE 3D contains three modules: *(1) Preprocessing:* using 2D foundation models to construct a featurized 3D point cloud (Section 2.1). *(2) Contextualized Representations:* produced by a PointTransformer-v3 (PTv3) (Wu et al., 2023) encoder that operates on the featurized point cloud to generate a contextualized 3D representation. The encoder is pre-trained with our novel SSL algorithm, 3D-JEPA (Section 2.2). *(3) Localization:* using a language-conditioned 3D object localization decoder that jointly predicts masks and bounding boxes (Section 2.3.1). The decoder is trained

from scratch, jointly with the 3D-JEPA initialized PTv3 encoder for the referential grounding task (Section 2.3.2).

## 2.1. Preprocessing: Lifting 2D Foundation Model Features into 3D Point Clouds

We begin by preprocessing the inputs (posed RGB-D images) by constructing a 3D pointcloud to encode geometry, and featurizing the pointcloud with off-the-shelf 2D foundation models to encode semantic information. We compute vision-only features from DINOv2 (Oquab et al., 2023), which are dense (patch-level). We also extract vision-language features from CLIP (Radford et al., 2021). Since CLIP features are global (i.e., one feature per input image) and not dense, we extract 2D instance masks from the input images using SAM (Kirillov et al., 2023) and compute per-mask CLIP features instead. These features are mapped back to the pixels containing the masks. We concatenate the CLIP and DINO features, along with a harmonic encoding of the RGB pixel intensities to obtain dense 2D feature maps. These feature maps are lifted to 3D similarly to (Jatavallabhula et al., 2023). We begin by unprojecting the RGB-D image to obtain a pointcloud, followed by voxelization (we use a voxel size of 5 cm across our experiments). We then compute a single feature per voxel by weighted averaging of all contained features. Weights are calculated using trilinear interpolation based on distance to voxel boundaries. This process results in a point cloud with "lifted" features, $\mathbf{PtC}_{\text{lift}} = \{(x_i, y_i, z_i, \mathbf{f}_i)\}_{i=1}^N$, where $\mathbf{f}_i \in \mathbb{R}^d$ is the feature vector of the $i^{th}$ point. This point cloud $\mathbf{PtC}_{\text{lift}}$ can be directly used for object localization. However, we find that refining the features further with an encoder, pre-trained with self-supervised learning (as described in Section 2.2), results in substantially better localization.

## 2.2. 3D-JEPA: Learning Contextualized Representations via SSL

3D-JEPA takes as input the lifted features $\mathbf{PtC}_{\text{lift}}$ from above, and learns a contextualized representation for the scene. Unlike the features in $\mathbf{PtC}_{\text{lift}}$, which are local to just the object/mask/patch, we seek learned features that attend to different parts of the scene to obtain representations that capture the entire context of the scene. This is analogous to learning contextualized embeddings (Devlin et al., 2019; Peters et al., 2018) as opposed to word-embeddings (Mikolov et al., 2013) in language.

To learn such a representation, we take inspiration from the Joint Embedding Predictive Architectures (JEPA) framework (Assran et al., 2023; Bardes et al., 2024) and develop 3D-JEPA. It is an encoder-predictor framework that performs masked prediction (He et al., 2022; Devlin et al., 2019) in the learned latent space. Concretely, let $\mathbf{PtC}$ be the input point cloud with some features (in our case, it

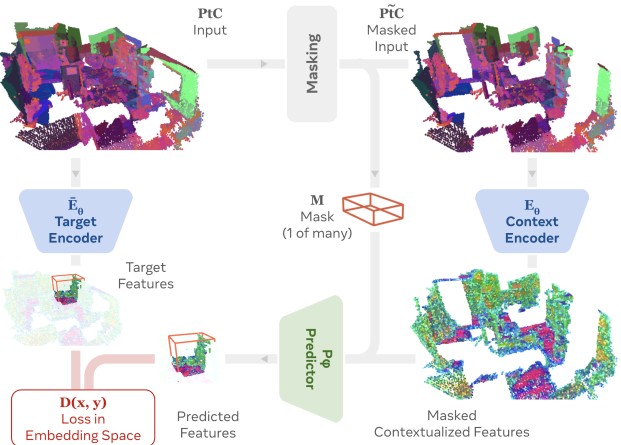

Figure 2: **3D-JEPA training framework**: The context encoder computes latent features from a masked point cloud. Subsequently, a predictor operates on these latent features to predict the features of masked regions. The target encoder has the same architecture as context encoder with weights being the exponentiated moving average of context encoder over course of training. The loss is computed per point in the embedding space and averaged across all points that were masked.

contains lifted 2D foundation features). 3D-JEPA trains two transformer models $E_\theta(\cdot)$ and $P_\phi(\cdot)$ using the following objective,

$$\min_{\theta, \phi} \left\| P_\phi \left( E_\theta(\mathbf{Pt\tilde{C}}), M \right) - \text{sg} \left( \bar{E}_\theta(\mathbf{PtC}) \right) \right\|, \quad (1)$$

where $\mathbf{Pt\tilde{C}}$ denotes a masked view of the point cloud, $M$ is a mask variable describing regions that were masked, and $\text{sg}(\cdot)$ is the stop-grad operator. Finally, $\bar{E}_\theta(\cdot)$ is the exponentiated moving average version of the encoder, which is important to prevent the representation from collapsing. The loss is computed per-point for all points in the masked region and then averaged. After training, $E_\theta(\cdot)$ is used as the contextualized representation.

Performing masked prediction in the latent space is advantageous over doing it in the ambient/input space for two reasons. First, unlike standard MAE in 2D vision, our input space is high-dimensional. In fact, since we use lifted 2D foundation features, our input dimension is equivalent to the ViT feature dimension. We found directly reconstructing such fine-grained and high-dimensional features to be difficult. Second, methods that leverage masked prediction in a teacher-student framework have produced the strongest results recently (Assran et al., 2023; Oquab et al., 2023).

To adapt the JEPA approach to 3D we developed a bounded radius-based masking strategy and efficient 3D native encoder and predictor architectures.

**Encoder and Predictor Architectures.** Unlike images or voxels that contain a regular grid structure, point clouds

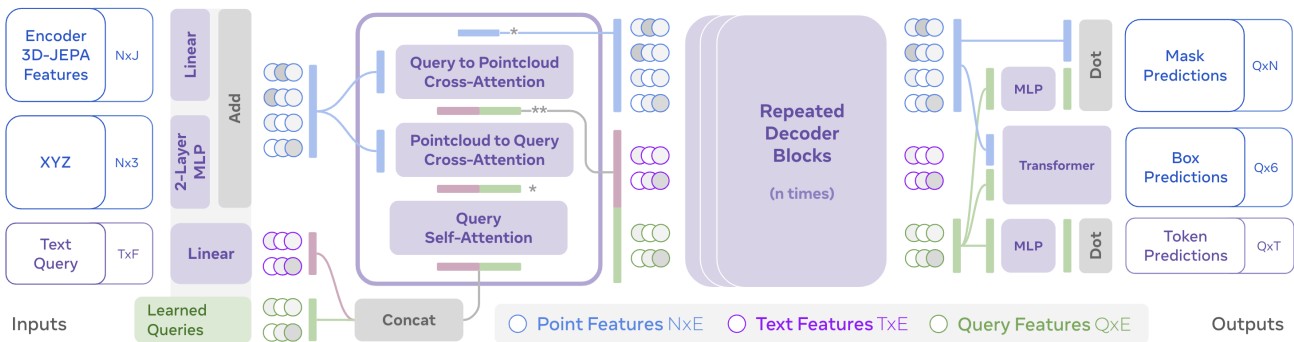

Figure 3: In our **language-conditioned 3D mask and bounding box decoder**, 3D-JEPA features are jointly processed with text and learned query embeddings by $n = 8$ decoder blocks and specialized prediction heads that generate mask, token, and box predictions. N is the number of points in the input pointcloud, T is the number of tokens in the input text, Q is the number of generated model queries, E is the decoder feature dimension, F is the input text feature dimension, J is the 3D JEPA feature dimension.

are order invariant and set-valued. U-Net (Çiçek et al., 2016), PointNet (Qi et al., 2016), DeepSets (Zaheer et al., 2017), and PointTransformer (Engel et al., 2020) have all emerged as promising architectures for point clouds. For our implementation of 3D-JEPA, we use Point Transformer v3 (PTv3) (Wu et al., 2023) for the encoder. In each layer, it first serializes the point cloud based on local proximity using bijective space filling curves. Subsequently, points are grouped together and they attend to each other within the group, which is loosely analogous to convolutions. For the predictor, we use a similar serialization step, but then use a standard transformer with sparse attention pattern. This allows for faster mixing of information from the start, due to lack of an explicit grouping. However, we found sparse attention to be crucial for training throughput and memory, as well as for training stability. See Appendix A.1 for further details about the architecture.

**Masking Patterns.** We found the choice of masking pattern to play a crucial role in the quality of representations learned, consistent with prior literature (Assran et al., 2023; Bardes et al., 2024). In particular, among the variants we tried, the serialized percent masking pattern shown in Figure 4 worked best. It contains two salient components: (1) masking out "regions" (points close in proximity) instead of random points to encourage better spatial understanding than simple local interpolation; and (2) masking out a percent of the scene instead of a fixed size allowing for training on point clouds of varying spatial sizes.

## 2.3. Object Localization from Referring Expressions

To solve the 3D referring expressions task, we design a language-conditioned 3D object localization decoder (Figure 3) that operates on the contextualized representations produced by our 3D-JEPA encoder (Section 2.2). This section describes the decoder architecture (Section 2.3.1) and end-to-end training procedure (Section 2.3.2).

### 2.3.1. LANGUAGE-CONDITIONED 3D DECODER

As illustrated in Figure 3, the decoder processes two inputs: the 3D-JEPA features $E_\theta(\mathbf{PtC}_{\text{lift}})$ and a text query $t$. These inputs are iteratively refined through a transformer and then fed into parallel prediction heads that generate 3D mask and bounding box predictions for all referenced objects. The details of this architecture are described below.

**Decoder Input Embedding.** We first project the 3D-JEPA $E_\theta(\mathbf{PtC}_{\text{lift}})$ representation into the model working dimension $E$ and add learned 3D positional embeddings. Similarly, we project the per-word CLIP (Radford et al., 2021) embeddings of the text query $t$. Next, we initialize a set of learnable object queries $Q$ and concatenate them with the language embeddings along the sequence dimension. In the following, we refer to the projected 3D-JEPA features as "point features" and the concatenated object queries and language tokens collectively as "queries."

**Decoder Blocks.** We apply a sequence of self-attention and cross-attention operations between the point features and queries. Specifically, each decoder module consists of three attention blocks: (1) a self-attention block that enables queries to refine their representations through mutual interaction, (2) a cross-attention block where queries extract relevant information from point features to enhance their contextual understanding, and (3) a final cross-attention block that updates the point features, ensuring they are informed by the refined queries. The first two blocks are standard from prior work (Vaswani et al., 2023; Carion et al., 2020; Cheng et al., 2021), while the final cross-attention block is inspired by (Jain et al., 2025), which demonstrates the impact of updating visual features for detection tasks in 3D. The decoder blocks are repeated for iterative refinement of the point features and queries. In LOCATE 3D, the number of decoder blocks are $n = 8$ and model dimension is $E = 768$. We observe a positive correlation between model scaling and performance, as discussed in Section 4.

**Decoder Heads.** Our decoder consists of three parallel prediction heads (Figure 7) that process the refined learned queries $Q$ independently as *object proposals*. Specifically, for each query, we jointly predict a 3D bounding box and 3D mask, using dedicated prediction heads. Additionally, following (Kamath et al., 2021), we predict an alignment matrix that grounds each query by determining which noun in the referring expression it corresponds to, if any.

The mask head follows the approach of (Cheng et al., 2021), by processing the queries with an MLP and computing a dot product with the point features to predict per-point mask logits. The text alignment head is composed of an MLP that receives the queries and directly predicts the alignment matrix. For bounding boxes, we developed a novel architecture (Figure 7). First, we concatenate linearly projected $x, y, z$ coordinates to the refined point features along the feature dimension. Then, we perform cross-attention between these concatenated point features and a linear projection of the refined queries. Finally, for each query, we use an MLP to regress bounding boxes.

### 2.3.2. TRAINING LOCATE 3D

LOCATE 3D trains the language-conditioned 3D decoder from scratch and fine-tunes the 3D-JEPA pretrained PTv3 encoder. It does so by using supervision signals from both object masks and bounding boxes to explicitly combine spatial constraints (boxes) and dense semantic supervision (masks); which leads to better localization performance (see experiments in Section 4.3).

Specifically, LOCATE 3D optimizes a composite loss function, which includes: (1) a mask loss, combining Dice and cross-entropy loss terms (Cheng et al., 2021); (2) a bounding box loss, composed of L1 distance and generalized IoU (Carion et al., 2020) terms; and (3) a text alignment focal loss with alpha balancing. Following (Carion et al., 2020), we define a matching cost and use Hungarian Matching to assign object query predictions to ground truth objects. We apply progressively weighted deep supervision at every decoder layer and maintain an Exponential Moving Average (EMA) of the model weights to use for evaluation and inference (Izmailov et al., 2018). In order to not destroy the pretrained features we use a stage-wise learning rate scheduler (Kumar et al., 2022); specifically we start by training the decoder with frozen encoder features, then fine-tune both jointly with a lower learning rate for the encoder. Further training details are provided in Appendix C.

## 3. LOCATE 3D DATASET Overview

LOCATE 3D DATASET (L3DD) is a new human-annotated referring expression dataset covering ScanNet (Dai et al., 2017), ScanNet++ (v1) (Yeshwanth et al., 2023), and ARK-itScenes (Dehghan et al., 2021). This section describes the dataset, and Section 4 discusses the impact of using L3DD to train our 3D referring expressions model.

### 3.1. Dataset Statistics

In total, our dataset contains 131,641 samples. Decomposed by scene dataset, L3DD contains:

1. **ScanNet**: 30,135 new language annotations covering 550 venues and 5,527 objects for training. 4,470 new language annotations covering 130 venues and 1038 objects for validation.
2. **ScanNet++**: 91,846 new language annotations covering 230 venues and 13,359 objects for training. 3,774 new language annotations covering 50 venues and 1,303 objects for validation.
3. **ARKitScenes**: 991 new language annotations covering 293 venues and 1,862 objects covering scenes used for pretraining. 425 new language annotations covering 93 venues and 460 objects for validation.

All validation split samples were validated by at least 1 human annotator. Over 80 percent of ARKitScenes and ScanNet++ validation split samples were validated at least three times, and samples were only included if a majority of annotators agreed that the sample was unambiguously correct.

### 3.2. Comparison with prior datasets

As shown in Table 9, L3DD significantly increases the scale of existing 3D RefExp data along two key axes when compared to prior data – language annotation quantity and scene coverage. Our language annotations approximately double the available quantity of training data, and after adjusting for the size of scenes, we approximately quintuple the number of size-adjusted venues with dense RefExp supervision. These annotations span multiple scene datasets, allowing principled study of scene generalization while holding the annotation process fixed. We show in Table 10 that this additional scene diversity is key to the value of L3DD as training data – holding the number of annotations fixed, training on ScanNet and ScanNet++ together significantly outperforms training only on ScanNet annotations (61.8% → 63.2% recall@0.25 on SR3D/NR3D/ScanRefer). Dataset visuals, collection procedure, and further analysis of L3DD is available in Appendix D.

## 4. Experiments and Analysis

In this section, we report results for our trained models. LOCATE 3D is trained and evaluated the standard 3D referential grounding benchmarks SR3D, NR3D (Achlioptas et al., 2020), and ScanRefer (Chen et al., 2020). LOCATE 3D+ additionally incorporates our newly collected L3DD

Table 1: **Results on 3D language grounding in 3D mesh and sensor point clouds (PC).** We evaluate top-1 accuracy on the validation set without any assumption of ground-truth proposals. We find that our model, LOCATE 3D, achieves state-of-the-art results across all three benchmarks for top-1 accuracy@25. *VLM-Baselines are evaluated on 2,000 samples per benchmark (6,000 total). **The localization decoder in LOCATE 3D was trained on training splits from the 3 benchmarks, while that for LOCATE 3D+ uses both the benchmark data as well as L3DD.

| Method | Joint Evaluation | | SR3D | | NR3D | | ScanRefer | |
|---|---|---|---|---|---|---|---|---|
| | Acc @25 | Acc @50 | Acc @25 | Acc @50 | Acc @25 | Acc @50 | Acc @25 | Acc @50 |
| *Mesh PC* | | | | | | | | |
| ReferIt3DNet Achlioptas et al. (2020) | 26.6 | - | 27.7 | - | 24.0 | - | 26.4 | 16.9 |
| ScanRefer (Chen et al., 2020) | - | - | - | - | - | - | 35.5 | 22.4 |
| InstanceRefer (Yuan et al., 2021) | 33.6 | - | 31.5 | - | 29.9 | - | 40.2 | 32.9 |
| LanguageRefer (Roh et al., 2022) | - | - | 39.5 | - | 28.6 | - | - | - |
| SAT-2D (Yang et al., 2021) | 37.14 | - | 35.4 | - | 31.7 | - | 44.5 | 30.1 |
| BUTD-DETR (Jain et al., 2021) | 50.28 | - | 52.1 | - | 43.3 | - | 52.2 | 39.8 |
| 3D-VisTA (Zhu et al., 2023) | 53.1 | 48.1 | 56.5 | 51.5 | 47.7 | 42.2 | 51.0 | 46.2 |
| PQ3D (Zhu et al., 2024) | 58.5 | **52.5** | 62.0 | **55.9** | 52.2 | **45.0** | 56.7 | **51.8** |
| *Sensor PC + Proposals from Mesh PC* | | | | | | | | |
| 3D-VisTA (Zhu et al., 2023) | 45.9 | 41.8 | 47.2 | 43.2 | 42.1 | 37.4 | 46.4 | 42.5 |
| ConcreteNet (Unal et al., 2024) | - | - | - | - | - | - | 56.12 | 49.50 |
| *Sensor PC* | | | | | | | | |
| 3D-LLM (Hong et al., 2023) | - | - | - | - | - | - | 30.3 | - |
| BUTD-DETR (Jain et al., 2021) | 40.7 | 26.6 | 43.3 | 28.9 | 32.2 | 19.4 | 42.2 | 27.9 |
| Llama VLM Baseline (Ours)* | 28.8 | 18.3 | 21.3 | 13.9 | 28.0 | 16.9 | 37.3 | 24.2 |
| GPT-4o VLM Baseline (Ours)* | 38.6 | 25.5 | 29.2 | 18.9 | 38.2 | 25.1 | 48.2 | 32.5 |
| **LOCATE 3D (Ours)** | **61.7** | 49.4 | **65.8** | 52.9 | **53.7** | 40.5 | **59.9** | 49.6 |
| **LOCATE 3D+ (Ours)** ** | **63.7** | 51.3 | **68.2** | 54.8 | **56.1** | 43.2 | **61.1** | 50.9 |

dataset for training. We evaluate on the validation split of the benchmarks and report top-1 accuracy without assuming ground-truth object proposals. Notably, we evaluate our methods on sensor point clouds obtained directly from lifting RGB-D observations, rather than the cleaned, post-processed point clouds sampled from mesh reconstructions provided by ScanNet. This choice better represents real-world deployment scenarios though it typically results in performance degradation due to sensor noise, missing regions, and registration errors, as discussed in (Jain et al., 2024). Section 4.1 compares Locate 3D to prior methods on standard benchmarks. Section 4.2 analyzes the impact of 3D-JEPA pre-training. Section 4.3 presents ablation studies on various components of our architecture, and Section 4.4 evaluates generalization capabilities on novel environments and robotic deployment.

### 4.1. How does LOCATE 3D compare to prior methods for 3D referential grounding benchmarks?

First, we study the performance of LOCATE 3D on three standard referential grounding benchmarks: SR3D, NR3D, and ScanRefer (Achlioptas et al., 2020; Chen et al., 2020). We compare with prior work and two vision-language model (VLM) baselines. The VLM baselines process the RGB-D

observations with a modular pipeline composed of three stages. In stage 1, a VLM – either Llama-3 (Meta AI, 2024) or GPT-4o (OpenAI, 2024) – is used to select a single 2D frame from the observation stream. In stage 2, a VLM selects an object in the selected frame, by choosing from 2D object masks generated with GroundingDINO (Liu et al., 2023) and SAM 2 (Ravi et al., 2024). In stage 3, the 2D mask for the selected object is propagated in time using SAM 2 (Ravi et al., 2024), and lifted to 3D using depth and camera information, to generate a predicted 3D bounding box. Further details of the Llama-3 and GPT-4o VLM baselines are provided in Appendix F.

We present the overall results in Table 1. Most prior work assumes access to refined meshes and mesh (object) region proposals at training and inference time. We instead choose to evaluate our model under more realistic conditions where only the sensor observation stream is available. Despite evaluating in this more stringent setting, our model (LOCATE 3D) achieves SoTA results, even when compared to prior work that operates under refined mesh point clouds. Furthermore, when also trained with our L3DD dataset (LOCATE 3D+), the model demonstrates even stronger performance, improving across all metrics while maintaining the same architecture and training methodology. We now discuss

Table 2: **Ablation study of input features and encoder configurations.** We compare different input modalities and encoder architectures. RGB refers to raw RGB features, while Concept-Fusion (CF) leverages CLIP+DINO-v2 features. We evaluate accuracy (@25 and @50 IoU) on the combined SR3D, NR3D, and ScanRefer evaluation sets. Results demonstrate that CF features consistently outperform RGB, and encoder initialization with 3D-JEPA yields the best performance. Note that in all experiments with an encoder, the encoder is fine-tuned with the decoder.

| Input | Encoder | Initialization | Acc@25 ↑ | Acc@50 ↑ |
|-------|---------|----------------|----------|----------|
| RGB | no | n/a | 28.9 | 17.4 |
| RGB | yes | PonderV2 | 42.2 | 30.1 |
| RGB | yes | random | 51.4 | 38.7 |
| CF | no | n/a | 53.9 | 39.7 |
| CF | yes | random | 59.8 | 47.0 |
| CF | yes | 3D-JEPA (frozen) | 56.2 | 44.0 |
| CF | yes | 3D-JEPA (**ours**) | **61.7** | **49.4** |

different components that lead to these performance gains.

### 4.2. Understanding the impact of 3D-JEPA

**Is 3D SSL necessary? Are lifted 2D foundation features sufficient?** We conduct a systematic ablation study examining three key aspects: (1) the choice of input features, (2) the role of utilizing an encoder architecture as opposed to simply training a decoder on lifted features and (3) the benefits of initializing the encoder with 3D-JEPA pre-training. For each configuration, we train the same type of decoder. The overall results are presented in Table 2.

We first examine the impact of input features, comparing raw RGB point clouds with lifted 2D foundation features. The results clearly demonstrate the importance of strong 2D foundation features, with CF showing a substantial improvement over RGB ($28.9\% \rightarrow 53.9\%$ Acc@25). When incorporating an encoder architecture, we observe interesting patterns. Even with random initialization, the encoder provides gains for both input types, though more pronounced for RGB ($28.9\% \rightarrow 51.4\%$) compared to CF features ($53.9\% \rightarrow 59.8\%$). Using a frozen 3D-JEPA encoder improves performance over the baseline CF features by 3% and 4% at Acc@25 and Acc@50 respectively, indicating that 3D-JEPA learns strong representations suitable for localization. Finally, we find that fine-tuning the 3D-JEPA encoder yields the best performance at 61.7%, underscoring the importance of SSL pre-training and task-specific finetuning.

**Does 3D-JEPA learn contextualized representations?** The key promise of SSL methods is to learn general purpose representations that are useful in many applications. 3D-JEPA accomplishes this by contextualizing, smoothing, and extracting the predictable features from the 2D foundational model inputs. We analyze the 3D-JEPA features through point-wise probing experiments for localization. We train a probe that operates on a text description (e.g., "chair near

entryway door") and features of a single 3D point to classify if the point satisfies the description or not. On this task, 3D-JEPA outperforms ConceptFusion 39% to 34%. On a relaxed "noun correctness" measure (e.g., is this point part of any "chair") 3D-JEPA outperforms ConceptFusion 73% to 66%. These results illustrate the power of 3D-JEPA features and their usefulness in many 3D vision domains.

Table 3: **Impact of 2D foundation features on LOCATE 3D**. We evaluate accuracy (@25 and @50 IoU) on the combined SR3D, NR3D, ScanRefer evaluation sets. LOCATE 3D is pre-trained and finetuned using different 2D foundation features. We find a clear trend: larger models (CLIP-L, SAM-H) as well as combining CLIP and DINO-v2 leads to better results. (SAM-) indicates masks computed with said model. (↑ indicates higher is better)

| 2D Foundation Features | Acc@25 ↑ | Acc@50 ↑ |
|------------------------|----------|----------|
| DINO-v2 | 53.7 | 39.4 |
| CLIP-B (MobileSAM) | 53.7 | 42.1 |
| CLIP-L (SAM-H) | 59.2 | 45.5 |
| CLIP-L (SAM-H) + DINOv2 (**ours**) | **61.7** | **49.4** |

### 4.3. LOCATE 3D ablations

**Benefits from advances in 2D foundation features.** Due to the massive amount of data available on the internet, 2D foundation models have been consistently improving. Do those advances also improve results in 3D? If so, this provides a powerful opportunity to leverage progress and advances there. To tease this apart, we trained variants of LOCATE 3D using different 2D foundation features. This goes through the full pipeline of lifting 2D features, pre-training the 3D backbone using said features, and end-to-end finetuning for 3D-RefExp using the localization decoder. The results are presented in Table 3. We find that using larger models (CLIP-L, SAM-H) improves results over smaller variants (CLIP-B, MobileSAM), suggesting benefits from scaling. Additionally, using CLIP-L and DINO-v2 improves results considerably over using CLIP-L alone. Thus, we find promising evidence that improvements in 2D foundation models translate to improved 3D object localization.

**What is the optimal decoder head architecture and supervision strategy?** Given our joint prediction task, we investigate two key design choices: (1) the type of supervision signal (mask-only, box-only, or both), and (2) the architecture of the bounding box prediction head. Our experiments in Table 5 show that mask-only supervision achieves moderate performance (55.4%) but lags behind our approach with a dedicated box head (61.7%). While post-processing these masks with DBSCAN helps address noisy predictions (58.4%), it still underperforms; particularly at higher IoUs (41.6% vs 49.4% at IoU@50) while introducing non-differentiable components into the pipeline. On the other extreme, box-only supervision leads to extremely poor per-

formance (0.3%), we hypothesize this is due to the lack of the more dense mask supervision signal. Finally, comparing box head architectures, we find that our transformer-based design significantly outperforms using a MLP (61.7% vs 35.6%), demonstrating the importance of properly incorporating spatial information through cross-attention.

**What is the impact of scaling the decoder?** We evaluate three different decoder sizes. As detailed in Appendix C.4, larger decoders consistently improve performance, particularly at higher IoUs. For all decoder sizes, 3D-JEPA pretraining provides a consistent 3-4% improvement in performance compared to randomly initializing the encoder.

**How to fine-tune 3D-JEPA?** We evaluate whether and how to fine-tune 3D in order to adapt it for the referential grounding task. We find that fine-tuning is necessary, as the frozen encoder underperforms (56.2% Acc@25), and a dedicate stage-wise learning rate schedule achieves the best results (61.7%), surpassing a baseline single-stage scheduler (59.5%); more details in Appendix C.4.

### 4.4. Evaluating LOCATE 3D in novel environments

**Performance on L3DD.** We evaluate LOCATE 3D's ability to generalize by testing it on the unseen scene dataset splits of our L3DD dataset (Section 3), which spans three scene datasets. We also evaluate on 412 samples on our held-out environment (FRE) used for robot testing. Even when trained only on ScanNet, LOCATE 3D demonstrates strong performance on L3DD's ScanNet++ and ARKitScenes. This is despite significant domain gaps. The **linguistic** distribution is different due to a different annotator pool and task instructions. The **scenes** were captured with different hardware by unique research efforts and span larger scenes and partial scans, both absent in ScanNet. And the **object** distribution is different, as ScanNet++ has denser open-vocabulary instance annotations. Notably, LOCATE 3D outperforms both baselines across most metrics, showcasing the robustness of our approach.

Our ablation studies reveal the key components enabling this strong generalization. First, replacing raw RGB inputs with lifted foundation features (CF) significantly improves cross-dataset performance across all benchmarks (SN++: 37.5% → 51.5%, ARKitScenes: 11.3% → 41.7%, FRE: 39.9% → 54.1%). The introduction of 3D-JEPA initialization (LOCATE 3D) further enhances generalization capabilities, boosting performance on SN++ to 56.7%, and ARKitScenes to 46.2%. Finally, in Table 7, we show that incorporating additional training data from L3DD (LOCATE 3D+) yields substantial improvements across all benchmarks (SN++: 56.7% → 83.9%, ARKitScenes: 46.2% → 57.6%, FRE: 52.0% → 73.5%). However, we observe that including in-domain training data from L3DD reduces the impact of 3D-JEPA pre-training on L3DD evaluations.

Table 4: **Generalization of LOCATE 3D.** We report accuracy @25 IoU. Using lifted 2D foundation features consistently improves results compared to RGB features, and 3D-JEPA pretraining further bolsters generalization. *GPT-4o VLM-Baseline evaluated on 2,000 samples SN++ and 6,000 samples for SN Joint.

| Method | Evaluation Dataset | | | |
|---|---|---|---|---|
| | ScanNet | LX3D | | |
| | Joint Eval | SN++ | ARKitScenes | FRE |
| **Baselines** | | | | |
| GPT-4o VLM | 37.6* | **60.5*** | 26.8 | 18.9 |
| CF + 3D-Decoder | 53.8 | 46.1 | 21.8 | 48.9 |
| **Ablations** | | | | |
| RGB, Random, SN | 51.3 | 37.5 | 11.3 | 39.9 |
| CF, Random, SN | 58.8 | 51.5 | 41.7 | **54.1** |
| **Ours** | | | | |
| LOCATE 3D | **61.7** | 56.7 | **46.2** | 52.0 |

**Deployment on Robot.** As outlined earlier, our model is capable of working with sensor streams and does not require human intervention at test time (e.g., for mesh refinement or instance segmentation). We deployed our LOCATE 3D model on a Spot robot in a held-out apartment scene. This scene is out-of-distribution by being a multi-room test apartment and is larger in size compared to the training data. The task was to navigate to a furniture object and pick up a "plush toy." Success was verified by grasping the toy. Our results show that LOCATE 3D achieved a success rate of 8/10 trials, outperforming baselines with a maximum success rate of 5.66/10 (see details in Table 11). Note that navigation and pick-up used pre-trained skills, while localization relied on our model. Additional details are provided in the supplementary video and Appendix E.1

### 4.5. Computational Analysis

**Run-time Analysis** For computational efficiency, we cache the environment's 2D features for each view, as well as the featurized point cloud. For ScanNet experiments, we compute this cache offline; and for robot experiments we compute it while doing the initial environment exploration phase. With this feature cache, a forward pass of our model takes 1 second for a scene with 100k feature points and utilizes 8 GB of VRAM on an A100 GPU.

**Limitations** We can utilize such caching because our benchmarks operate under static (ScanNet) or quasi-static (robot) environments. Extending our approach to dynamic scenes would require real-time 2D feature computation and continuous updates to the featurized environment. We believe the former is a matter of engineering, while the latter is an active research area, explored by methods like Lifelong LERF (Rashid et al., 2024)

# 5. Related Work

**Self Supervised Learning (SSL)**, and more broadly representation learning for 3D data like point clouds, is relatively under-explored compared to 2D vision (Fei et al., 2023). Among the few explorations, most have focused on learning representations for individual objects using object-level point clouds (Sauder and Sievers, 2019; Pang et al., 2022; Zhang et al., 2022). PointMAE (Pang et al., 2022; Zhang et al., 2022) and PointBERT (Yu et al., 2022) use masked modeling techniques for object-level point clouds and show that the resulting representations can be effective for tasks like classification and segmentation. They differ crucially from our work on two fronts. First, our work is focused on scene-level point clouds which are significantly larger compared to single object point clouds used in prior work. Second, we operate on point clouds with high feature dimensions, as opposed to RGB point clouds in prior work. In our experiments we found that a naive instantiation of masked modeling is ineffective due to these factors. PointCLIP (Zhang et al., 2021a) leverages 2D foundation features for CLIP, but again looks only at object-level point clouds. Prior works have also explored contrastive learning methods (Xie et al., 2020; Zhang et al., 2021b), which encourage representations to be invariant to different types of transformations. Other approaches have explored distilling 2D features directly to 3D, by differentiable rendering ( (Kobayashi et al., 2022); (Cao et al., 2025)) or constrastive learning (Peng et al., 2023). Ponder-v2 (Zhu et al., 2025) proposes an SSL pretext task based on differentiable rendering into 2D. We refer readers to a recent survey on SSL for point clouds for additional references (Fei et al., 2023).

**3D Referential Grounding:** Recent years have seen renewed interest in tasks at the intersection of perception and language, such as visual grounding (Zhang et al., 2023; Liu et al., 2023; Ren et al., 2024) and visual question answering (Antol et al., 2015; Sermanet et al., 2023; Majumdar et al., 2024). Of particular relevance to this work are 3D referential grounding benchmarks, such as SR3D, NR3D, and ScanRefer (Achlioptas et al., 2020), which require localizing objects mentioned in a language utterance using observations of a 3D scene. We note that the original instantiations of these benchmarks provide access to ground-truth bounding boxes of all objects in the scenes as input (including at test time), and the task is to select the correct bounding box. In contrast, LOCATE 3D operates directly on sensor observation, without requiring any annotations. Few prior works operate under this setting.

Current approaches for 3D grounding fall under two classes of methods. First are end-to-end models designed explicitly for this task, that operate in one of two modes. In the *grounding-by-detection* setting, models are trained to directly the bounding box corresponding to the object referred to in the query (Luo et al., 2022; Jain et al., 2021). In the *grounding-by-selection* setting, a pretrained vision backbone extracts region proposals (bounding boxes or segmentation masks), and models are trained to select the best proposal that matches the query (Chen et al., 2022; Zhu et al., 2023; 2024; Unal et al., 2024). These approaches have so far not been demonstrated on sensor point clouds or scaled up to multiple datasets, with the sole exception being ConcreteNet (Unal et al., 2024), which trains a model to rank proposals from scratch. The primary limitation with this class of approaches is in the requirement of an external region proposal mechanism, which is often difficult in 3D, and are prone to failures.

The second class of methods aim to leverage 2D foundation models (VLMs) for 3D tasks. While direct lifting of 2D features has shown promising performance on localizing objects from simple noun phrases (Jatavallabhula et al., 2023; Kobayashi et al., 2022; Wang et al., 2024), this does not translate to 3D referential grounding. Thefore, approaches have either relied on using stronger models such as VLMs (Yang et al., 2023a; Xu et al., 2024), or by finetuning a separate head for the task (Hong et al., 2023; Huang et al., 2024). While these methods inherit the strong generalization capabilities of foundation models, they rely on tools for grounding and inherit their weaknesses. In the broader context, one could interpret LOCATE 3D as a powerful tool that such agents can leverage to perform better in the future.

# 6. Conclusion

In this work, we introduced LOCATE 3D, a model for localizing objects in 3D from textual referring expressions. Our approach leverages 3D-JEPA, a novel self-supervised learning method for point clouds. It first projects features from 2D foundation models (like SAM, CLIP, DINO) into 3D point clouds. Subsequently, it performs SSL on top of these lifted features with a pretext task of masked prediction in latent space - i.e. predict latent features of masked regions using the remainder of the scene. We showed that this enables the learning of contextualized representations – i.e. features of a particular point take into account the whole scene. We also showed that a backbone pre-trained with 3D-JEPA can be effectively finetuned for the task of RefExp, using a mask-and-box decoder, that ultimately results in LOCATE 3D. We show that LOCATE 3D achieves state-of-the-art results in standard RefExp benchmarks. Furthermore, unlike prior work, LOCATE 3D only requires sensor point clouds, making it suitable for applications like robotics and smart glasses.

## Impact Statement

This paper presents work whose goal is to advance the field of Machine Learning. There are many potential societal consequences of our work, none which we feel must be specifically highlighted here.

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

# A. Model Details

## A.1. Pre-training

We experiment with several masking strategies to achieve the best pre-training representation. Two strategies for computing masks were percent-based and fixed number of points. In percent-based masking, we mask a percentage of the scene. E.g. we mask 8% of the scene. This means that depending on the size of the pointcloud that range from 1k points to 200k points, the number of points in the masks varies widely. In fixed number of points masks, we keep the number of points in every mask constant and vary the number of masks based on the size of the pointcloud to ensure we cover similar portion of the scene. We found percent-based masking to perform best.

Additionally, we experiment with two methods for constructing masks: radius-based and serialized-curve based. In radius-based masking, for a mask of size $x$, we randomly select a point in the scene and select the $x$ closest points based on the Cartesian coordinates. In serialized-curve masking, we first order the points based using the serialization methods. We then mask contiguous blocks in this serialized sequence. We find that using serialized-curve based masking works best as the mask regions have more diversity than the circular radius-based masking.

We evaluate the masking strategies using a frozen representation element-wise probe. We show the performance of the element-wise probe on the referring expression benchmark throughout the pre-training in Figure 5.

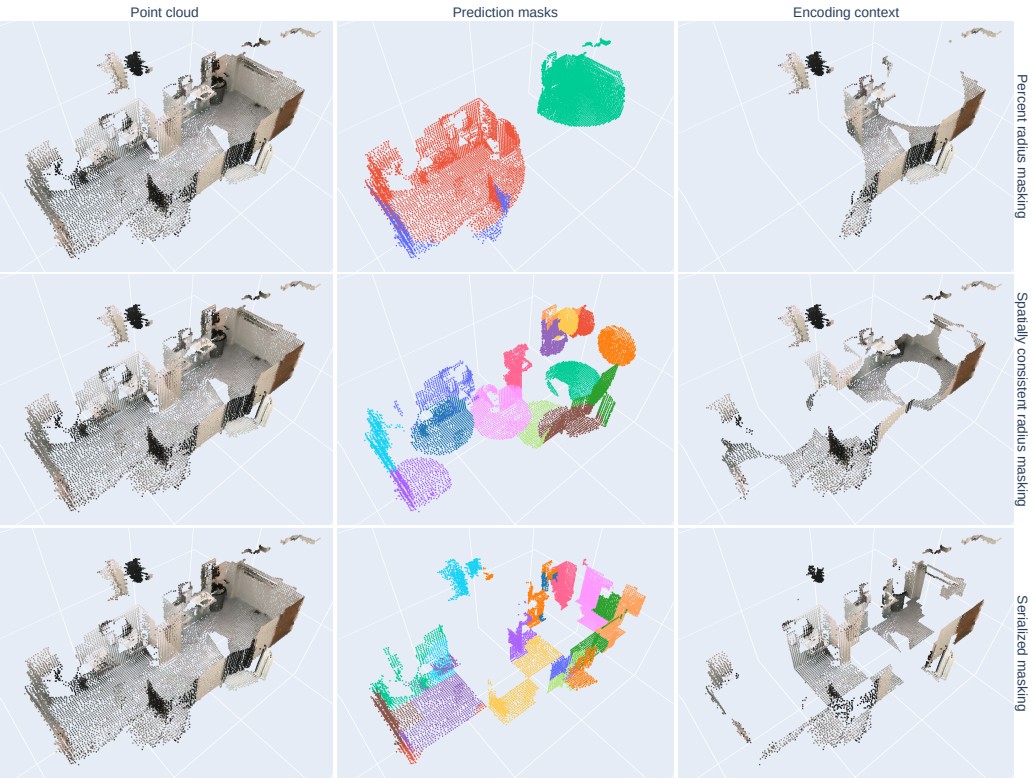

Figure 4: Overview of different masking types

## A.2. Encoder architecture

The main backbone of the encoder architecture is the PointTransformerV3 archicteture. For input to the transformer, we have CLIP, DINO and RGB features. For RGB, we harmonically embed the RGB features before using a small MLP to process the features. Given that we run CLIP on each of the SAM masks for an image, some pixels are not contained in any masks and have no CLIP features. We use a learnable parameter to represent these points without CLIP features. Finally, we concatenate the MLP processed harmonically embed RGB, learnable parameter masked CLIP features, and the DINO features together. We then tokenize these points with a sparse sub-manifold stem convolution before using the

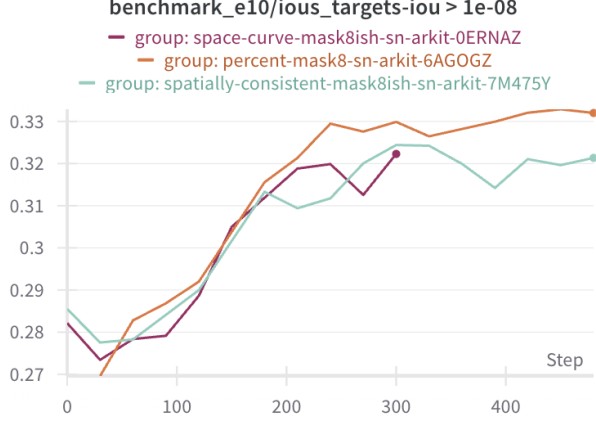

Figure 5: Element-wise probe results during pre-training with different masking strategies

PointTransformerV3 architecture.

### A.3. Predictor Architecture

The predictor architecture takes the encoded context along with a learnable parameter mask tokens to represent the points to predict as well as global registers. These inputs are process in a transformer with a block sparse attention pattern and rotary positional embedding. The rotary positional embeddings use the continuous Cartesian coordinates of the points to rotate sub-section of the vector for each x, y, and z directions. The sparse attention mask is constructed so the mask tokens can all attend to all of the encoded context, the encoded context can diagonally self-attend. An example sparse attention mask is shown in Figure 6.

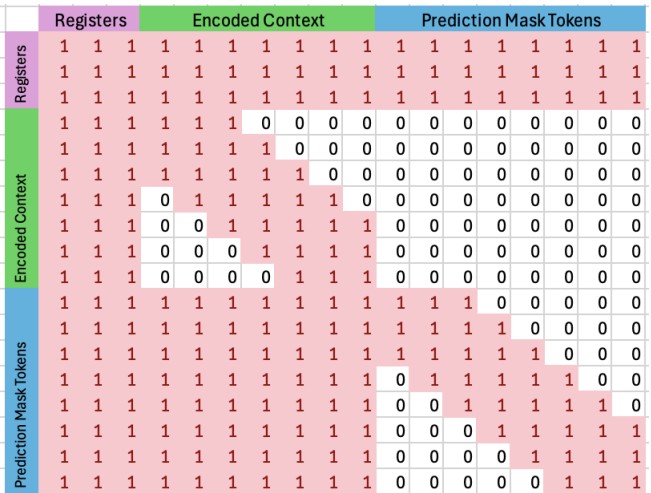

Figure 6: Left: Block sparse attention pattern

## B. Object Localization

In this section, we discuss details regarding our Language-Conditioned 3D Decoder and the end-to-end training for referential grounding.

## B.1. Decoder architecture

In Figure 7, we provide detailed illustrations for our token prediction head (top left), mask prediction head (top right), and bounding box prediction head (bottom).

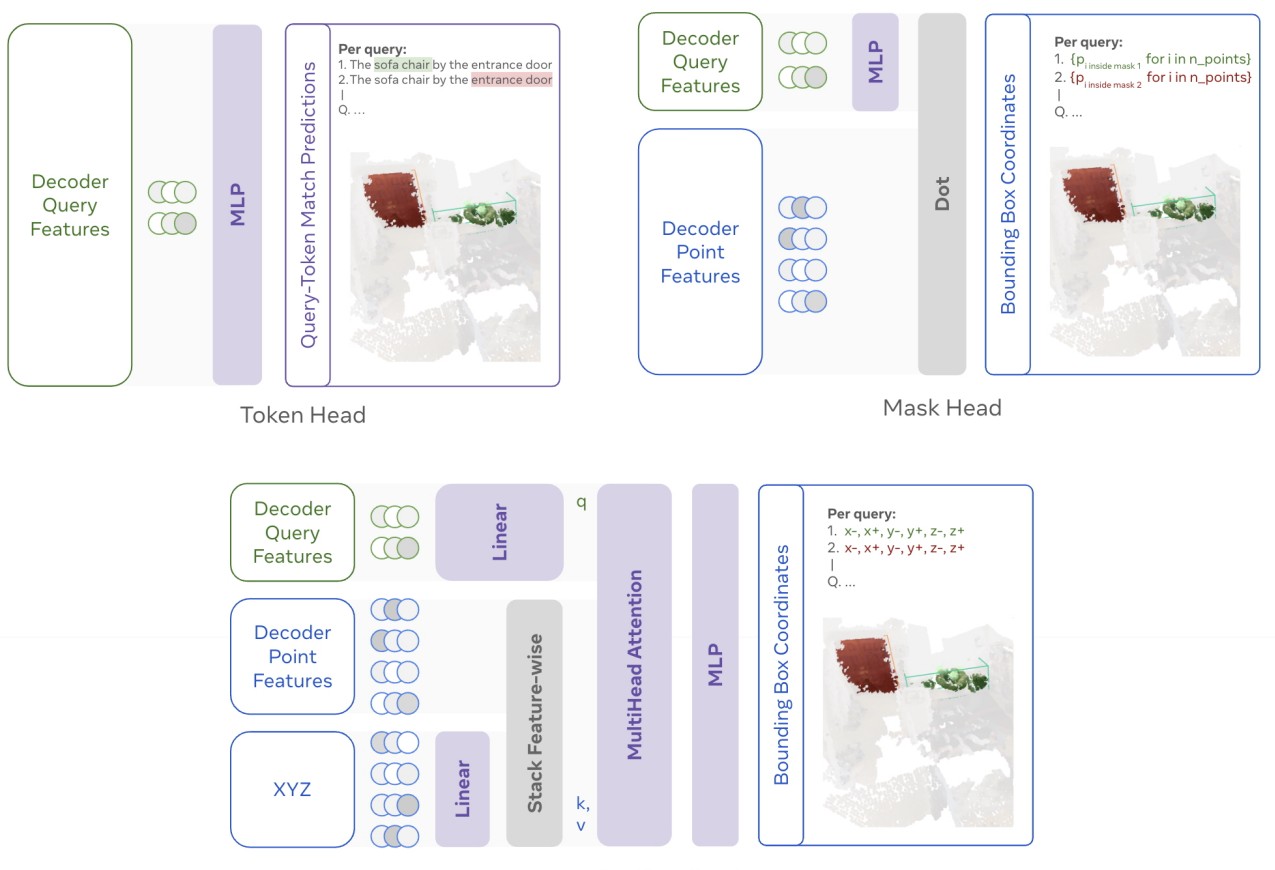

Figure 7: **Prediction heads of the language-conditioned decoder** (Top Left) Our Token Prediciton Head (Top Right) Our Mask Head (Bottom) Our bounding Head

## B.2. Decoder Ablations

We present the results of additional experiments on decoder supervision objectives and prediction head architectures in Table 5 and decoder size in Table 6.

## C. LOCATE 3D training

### C.1. Details of training

LOCATE 3D is optimized using AdamW (Loshchilov and Hutter, 2019) with parameters $\beta_1 = 0.9$, $\beta_2 = 0.999$, weight decay of $0.01$ and using a learning rate scheduler as described in Appendix C.2. We optimize the following loss function:

$$\mathcal{L} = \lambda_{\text{dice}}\mathcal{L}_{\text{dice}} + \lambda_{\text{ce}}\mathcal{L}_{\text{ce}} + \lambda_{\text{box}}\mathcal{L}_{\text{box}} + \lambda_{\text{giou}}\mathcal{L}_{\text{giou}} + \lambda_{\text{align}}\mathcal{L}_{\text{align}}$$

Table 5: **Ablation study on decoder supervision and bounding box prediction head architectures.** We evaluate accuracy (@25 and @50 IoU) on the combined SR3D, NR3D, and ScanRefer evaluation sets (Joint ScanNet). We observe that without box supervision, DBSCAN can be used remove outliers and improve accuracy. However, using box supervision with our transformer architecture leads to the best performance and removes the reliance on additional post-processing (with DBSCAN).

| Supervision | | Architecture | Joint ScanNet | |
|---|---|---|---|---|
| Mask | Box | Box Head | Acc@25 | Acc@50 |
| no | yes | Transformer | 0.3 | 0 |
| no | yes | MLP | 29.4 | 10.0 |
| yes | no | Naive | 55.4 | 39.1 |
| yes | no | DBSCAN | 58.4 | 41.6 |
| yes | yes | MLP | 35.6 | 13.5 |
| yes | yes | Transformer **(ours)** | **61.7** | **49.4** |

Table 6: **Impact of decoder size on performance.** We evaluate accuracy (@25 and @50 IoU) on the Joint ScanNet benchmark across different decoder sizes (Small, Base, and Large). Larger decoders consistently improve accuracy, while 3D-JEPA pre-training provides a stable performance boost across all scales.

| Initialization | Decoder Size | Acc@25 $\uparrow$ | Acc@50 $\uparrow$s |
|---|---|---|---|
| Random | Small | 54.4 | 39.0 |
| Random | Base | 57.6 | 44.5 |
| Random | Large | 58.8 | 46.3 |
| 3D-JEPA | Small | 58.1 | 43.3 |
| 3D-JEPA | Base | 60.6 | 47.6 |
| 3D-JEPA | Large | **61.7** | **49.4** |

With:

$$\lambda_{ce} = 4.0 \quad \text{(Class weight)}$$
$$\lambda_{mask} = 6.0 \quad \text{(Mask cross entropy weight)}$$
$$\lambda_{dice} = 4.0 \quad \text{(Mask dice weight)}$$
$$\lambda_{box} = 1.0 \quad \text{(Bounding box L1 weight)}$$
$$\lambda_{giou} = 1.0 \quad \text{(Bounding box GIoU weight)}$$

## C.2. Training Schedule Details

Fine-tuning a pre-trained encoder alongside a randomly initialized decoder presents significant challenges. The primary issue stems from the decoder's initially poor gradients potentially destabilizing the valuable pre-trained representations in the encoder. To address this, we designed a training schedule that carefully balances the learning dynamics of both components while preserving the pre-trained features.

Our approach (Figure 8 follows a progressive training strategy where components are introduced gradually into the optimization process. The schedule consists of four primary phases: initial decoder training with a frozen encoder, a transition period introducing encoder fine-tuning at a reduced rate, an encoder adaptation phase, and finally joint optimization with component-specific learning rates. Throughout training, we maintain lower learning rates for the encoder (0.5× the decoder rate) and implement smooth transitions between phases to ensure stability.

The complete schedule spans 40 epochs, with the first 17 epochs dedicated to the careful initialization and adaptation phases.

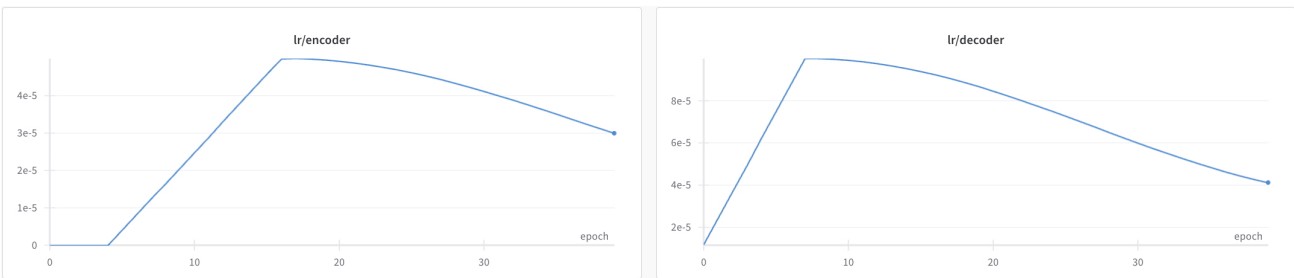

Figure 8: **Learning rate schedule for encoder and decoder.** Fine-tuning a pre-trained encoder alongside a randomly initialized decoder requires careful balancing to prevent unstable gradients from disrupting pre-trained representations. Our progressive training schedule spans 40 epochs, with an initial phase freezing the encoder, followed by gradual adaptation and joint optimization. The encoder (left) maintains a lower learning rate (0.5× the decoder rate) with smooth transitions, ensuring stable fine-tuning and improved performance.

### C.3. Finding the Right Sensor Supervision for LOCATE 3D

A key challenge in training LOCATE 3D is the mismatch between our input data (raw sensor information) and the available annotations (clean mesh pointclouds) in ScanNet. While ScanNet provides RGBD+pose trajectories, allowing us to reconstruct uncleaned 3D pointclouds via SLAM that better match real-world conditions, the question remains: how do we effectively supervise a model operating on noisy sensor data using annotations from clean meshes?

We explored two approaches for bridging this gap:

- **Mesh-space supervision:** Transfer model predictions from sensor pointcloud to mesh pointcloud by assigning each mesh point the distance-weighted average of predictions from its k-nearest neighbors (k=8) in the sensor pointcloud. Supervision occurs in mesh space.

- **Sensor-space supervision:** Transfer mesh annotations to sensor pointcloud by assigning each sensed point the distance-weighted average of ground truth labels from its k-nearest neighbors ($k = 8$) in the mesh. Supervision occurs directly in sensor space.

Approach (2) proved superior, delivering both practical benefits in terms of efficiency (single transfer operation vs. one per decoder layer) and significantly better performance. The key insight came from analyzing failure cases: in approach (1), about 50% of points were effectively ignored during training because they had no close neighbors in the clean mesh. These ignored points were precisely the sensor artifacts and noise that the model needs to learn to handle in real-world deployments. By supervising in sensor space, we force the model to process all input points, including challenging cases like sensor noise and artifacts.

### C.4. Training ablations

**How to fine-tune 3D-JEPA?** Given a pre-trained 3D-JEPA encoder and a randomly initialized decoder, we investigate different strategies for jointly fine-tuning these components. Specifically, we compare four configurations: (1) a randomly initialized encoder trained end-to-end with a naive learning rate schedule (warmup + cosine decay with equal learning rates for both components), (2) a frozen pre-trained 3D-JEPA encoder, (3) a pre-trained 3D-JEPA encoder fine-tuned with the naive schedule, and (4) a pre-trained 3D-JEPA encoder trained with our stage-wise training Appendix C.2.

The experiment results are detailed in Table 8, the frozen variant performs moderately (56.2% Acc@25) but lags behind the random initialization where the model is fine-tuned end-to-end (59.8%); naive fine-tuning of the 3D-JEPA initialized encoder marginally improves results (59.5%), and our stage-wise schedule achieves the best result 61.7%.

**What is the impact of scaling the decoder?** We investigate the effect of decoder scaling by evaluating three model sizes (Small, Base, and Large) with both random initialization and 3D-JEPA pre-training. As shown in Table 6, increasing decoder size consistently improves performance, with each scaling step (Small→Base→Large) yielding approximately

Table 7: **Impact of LX3D train data.** We report accuracy @25 IoU. ARKitScenes column contains both pretrain and val split as we saw no significant difference when split up. Adding LX3D training data significantly improves performance on unseen LX3D scenes. 3D-JEPA's impact on LX3D evaluations is reduced when adding LX3D train data. ± shows standard error.

| Training Data | Evaluation Dataset | | | | |
|---|---|---|---|---|---|
| | Existing SN Benchmarks | | LX3D | | |
| | ScanRefer | NR3D | SN++ | ARKitScenes | FRE |
| ScanNet, JEPA (LOCATE 3D) | 59.9 ± 0.58 | 53.7 ± 0.51 | 56.7 ± 0.74 | 46.2 ± 1.32 | 52.0 ± 2.46 |
| ScanNet, Random Init | 56.7 ± 0.58 | 51.6 ± 0.51 | 51.5 ± 0.75 | 41.7 ± 1.31 | 54.1 ± 2.46 |
| ScanNet + LX3D, JEPA (LOCATE 3D+) | **61.1 ± 0.57** | **56.1 ± 0.51** | **83.9 ± 0.55** | 57.6 ± 1.31 | 73.5 ± 2.17 |
| ScanNet + LX3D, Random Init | 61.0 ± 0.58 | 54.9 ± 0.51 | 83.5 ± 0.56 | **59.3 ± 1.31** | **73.8 ± 2.17** |

1-3% absolute improvement in Acc@25 and 2-5% in Acc@50 on our joint ScanNet evaluation. The larger gains at higher IoU thresholds suggest that increased model capacity particularly benefits precise object localization. Notably, the benefit of 3D-JEPA pre-training remains relatively stable across all scales, providing an ∼3-4% improvement over random initialization for each decoder size. These results suggest that model capacity and initialization quality contribute independently to performance, with both larger decoders and better pre-training being beneficial for the 3D localization task.

### C.5. Visualizing LOCATE 3D

Figure 9 visualizes the transformation of point cloud features as they pass through the model's encoder and decoder. The left column shows the original RGB scenes, while the middle and right columns present PCA-reduced embeddings of the 3D point features extracted from the fine-tuned 3D-JEPA encoder and the decoder's refined output, respectively.

The encoder processes the lifted point cloud representation, capturing global scene context and producing initial feature embeddings. The decoder then iteratively refines these representations through a series of self-attention and cross-attention operations, incorporating language-conditioned object queries. As seen in the PCA visualization, the encoder seems to learn more general semantic features that *look* like a zero-shot segmentation; the decoder enhances the feature distinctiveness around the referenced objects, leading to sharper and more localized embeddings.

Table 8: **Impact of learning rate schedule on model performance.** We evaluate accuracy (@25 and @50 IoU) on the combined SR3D, NR3D, and ScanRefer evaluation sets. The stage-wise schedule prevents catastrophic forgetting of pre-trained features to enable effective fine-tuning.

| Encoder Initialization | LR Schedule | Acc@25 ↑ | Acc@50 ↑ |
|---|---|---|---|
| Random | Naive | 58.8 | 46.3 |
| Random | Stage-wise | 59.8 | 47.0 |
| 3D-JEPA | Frozen | 56.2 | 43.7 |
| 3D-JEPA | Naive | 59.5 | 47.0 |
| 3D-JEPA | Stage-wise (**ours**) | **61.7** | **49.4** |

## D. Annotation Details

### D.1. Collection Procedure

We collect data using an off-the-shelf annotation interface which allows for selecting rectangular regions on 2D videos and associating them with text. We project 3D instance masks into 2D and show them alongside the RGB camera view. For ScanNet (Dai et al., 2017) and ScanNet++ (Yeshwanth et al., 2023), we use ground truth instance masks provided by the dataset. For ARKitScenes (Dehghan et al., 2021) and our robot trial environment, we use instance masks produced by SAMPro3D (?). Annotators have the ability to conglomerate multiple instance masks into a single object – this proved

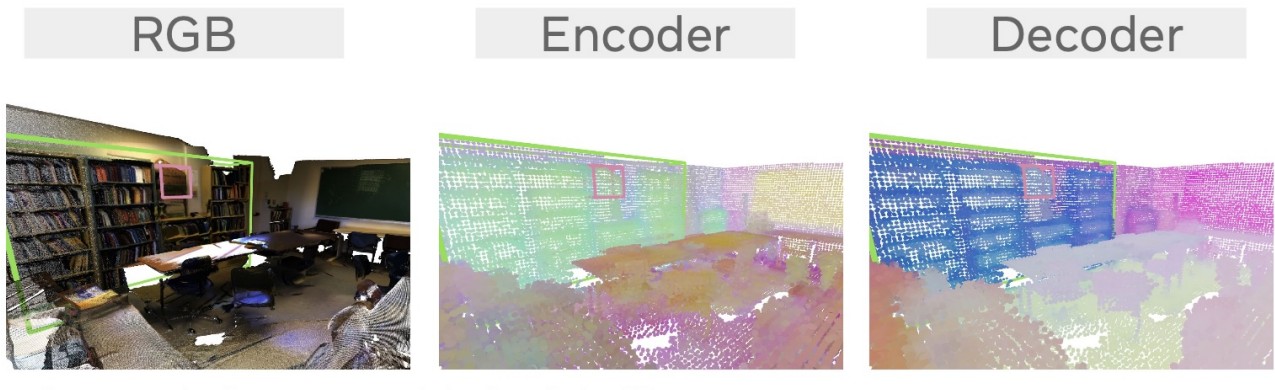

"Locate the box on top of the bookshelf"

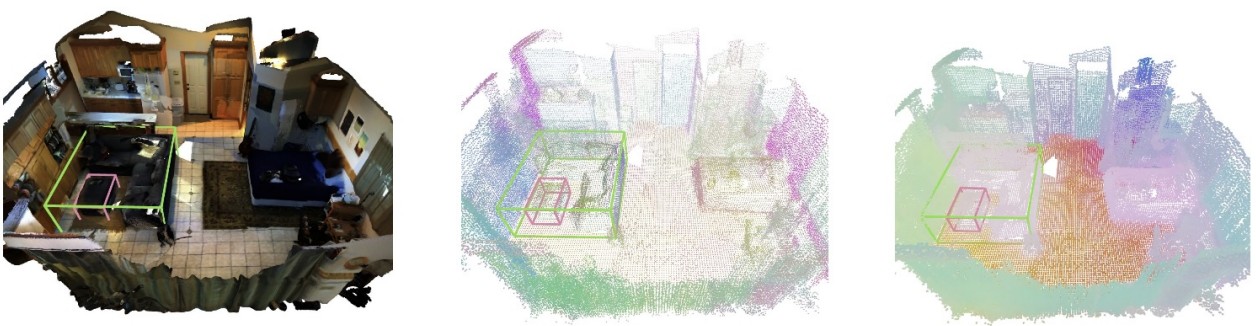

"Locate the coffee table in front of the L-shaped couch"

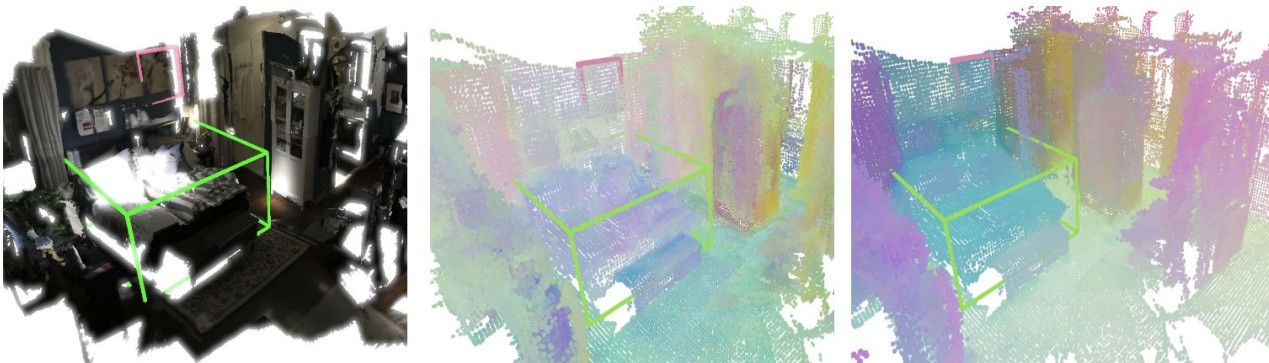

"Locate the far right painting above the bed"

Figure 9: **Visualization of point cloud features before and after decoding.** We compute the PCA of the fine-tuned 3D-JEPA encoder's point features (middle column) and the refined point features after decoding (right column). The RGB images (left column) provide reference scenes, while the PCA projections reveal that the encoder seems to learn smooth semantic features while the decoder learns sharper localized features.

necessary for SAMPro3D segmentations, which tended to break large objects into many subparts. This is in line with ScanNet's instance segmentation procedure in which annotators stitch together an automatically-generated oversegmentation of a 3D scene (Dai et al., 2017).

In addition to grounding the target object, annotators were asked to provide object grounding for all other nouns in the phrase they provided. For SAMPro3D masks, annotators could select any object they wished as a target. In later iterations of the task, for ScanNet and ScanNet++, we highlighted a mask in white with at least 1 distractor to serve as the target object for description. This is in line with prior literature which observed that the 3D RefExp task is most challenging in the high-distractor case (Chen et al., 2020). We later rely on an LLM (Llama 3.2-90B) to identify which object is the target.

While we found that annotations on ScanNet and ScanNet++ were relatively reliable from a first pass, we observed considerable mask/bounding box noise when collecting annotations on SAMPro3D instance masks. To address this, we created a separate validation task. In this task, annotators saw a video similar to the video provided in the generation task, but only the selected masks for a particular language sample were highlighted. The RefExp description was also shown. This improved the quality of SAMPro3D samples substantially, but it also drastically slowed the collection rate. As such, we do not use ARKitScenes for supervised training.

Additionally, as we had the validation task set up, we had annotators validate every eval sample at least once (with over 80 percent of ScanNet++ and ARKitScenes samples receiving 3+ validation labels). Annotators were instructed to only keep accept samples which were unambiguously correct, and we only samples which a majority of annotators marked as trustworthy.

### D.2. Annotation setup

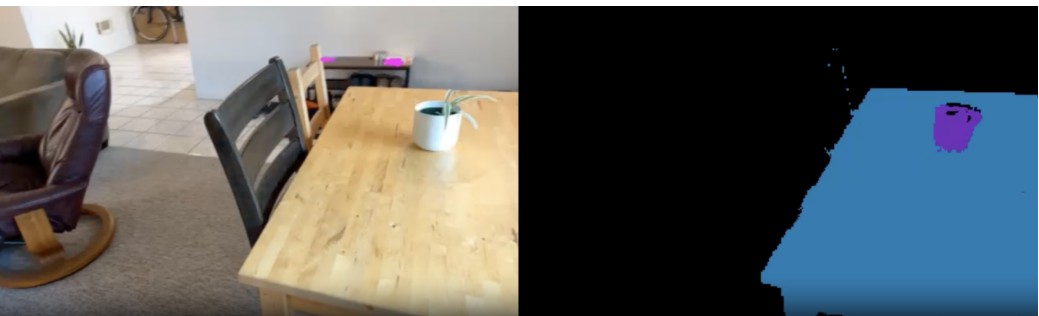

Figure 10: An example sample from the RefExp task. This sample is taken from the ScanNet++ (Yeshwanth et al., 2023) split of L3DD. The model receives the camera input (RGB shown on left) with additional depth and pose information and must produce the object masks (shown on the right). The text conditioning is "indoor plant on the table."

Our annotation setup largely mirrors ScanRefer (Chen et al., 2020) – annotators simply describe objects in a one-step fashion rather than the two-player game format of NR3D (Achlioptas et al., 2020). We use a 2D video interface for the task, projecting 3D instance masks to 2D. Annotators can write full referring expressions and associate particular instance masks with particular tokens in the expressions. They may also agglomerate two or more instances into one.

When ground-truth instance masks are available (as in ScanNet and ScanNet++), we use these as the basis for annotation. When they are not available (as in ARKitScenes and the demo FRE apartment scene), we generate instance masks using SAMPro3D (**?**).

We implemented two versions of the task. Initially, we aimed to simplify the task as much as possible so that a high quantity of annotations on SAMPro3D masks would be feasible. As it was challenging to achieve both speed and quality with this approach, we switched to a more challenging task to annotate existing instance masks.
**Version 1:** Annotators select only two objects and may choose any object in the scene as a target.
**Version 2:** Annotators select an arbitrary number of objects. When comprehensive semantic instance annotations are available (as in ScanNet/ScanNet++), we highlight an object with at least 1 distractor as a prescribed target.

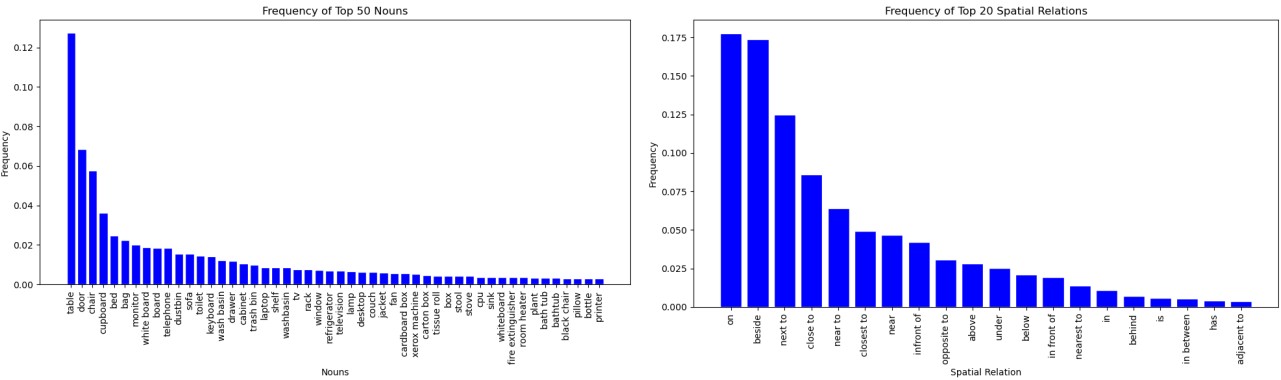

Figure 11: Histogram of most common nouns and spatial relations in L3DD. Computed over 5000 samples using Llama 3.1-8B. Left: Top nouns, Right: Top spatial relations.

## D.3. Comparison with existing datasets

We show in our ablations that mask supervision is crucial for training. Voxel coverage is the number of 5cm$^3$ voxels occupied by at least one point in the sensor pointclouds of these datasets. When multiple scans are available of a single venue, we only use the voxel occupancy of the first scan so as to ensure this acts as a measure of size-adjusted *venue* coverage. Finally, we compute object coverage by enumerating the number of unique object instance masks covered by each dataset. For the case of the ARKitScenes split of our data, we divide the number of SAMPro3D segments by the average number of segments per object instance (2.25). Note that this object count includes both grounded targets and grounded anchors.

We showed in Table 5 that mask supervision is crucial for high-quality training data, so ARKitSceneRefer's commendable object diversity is not easily applicable to our method. Of the other three datasets with grounded anchors and mask coverage, there is significant overlap in instance coverage; the three together cover 29,857 instances. All of L3DD's 14,662 new instances in ScanNet++ have not yet been covered by a RefExp dataset.

Additional dataset statistics (*starred values were computed using Llama 3.1-8B instruct(Meta AI, 2024) over 5000 samples)

- Average query length (in words): 7.55

- Average number of grounded instances per query: 2.28

- Vocabulary size: 3058

- Percent of samples containing a reference to color: 42.6*

- Percent of samples containing a reference to object shape: 38.22*

Table 9: A comparison of LOCATE 3D DATASET with other 3D RefExp datasets. L3DD advances the state-of-the-art by covering multiple scene datasets and a large number of unique venues with mask-grounded 3D referring expressions.

| Dataset | Scene Datasets | Objects | Expressions | Venues | Voxels Covered | Human-generated | Masks | Grounded Anchors |
|---|---|---|---|---|---|---|---|---|
| ScanEnts3D-SR3D (Abdelreheem et al., 2024) | 1 | 16,797 | 83,572 | 613 | $26.1 \times 10^6$ | ✗ | ✓ | ✓ |
| ScanEnts3D-NR3D (Abdelreheem et al., 2024) | 1 | 14,710 | 41,503 | 641 | $26.9 \times 10^6$ | ✓ | ✓ | ✓ |
| ScanEnts3D-ScanRefer (Abdelreheem et al., 2024) | 1 | 18,989 | 51,583 | 800 | $28.8 \times 10^6$ | ✓ | ✓ | ✓ |
| ARKitSceneRefer (Kato et al., 2023) | 1 | 15,553 | 15,553 | 1,605 | $187.8 \times 10^6$ | ✓ | ✗ | ✗ |
| LOCATE 3D DATASET (ours) | **3** | **23,549** | **131,641** | 1,346 | $123.7 \times 10^6$ | ✓ | ✓ | ✓ |

## D.4. Why does the additional training data help?

We ran an additional experiment in which we find that scene diversity is a key factor in L3DD improvements. Specifically, we compare two conditions: (1) ScanNet training data + 30K L3DD samples also from ScanNet and (2) ScanNet data

| Train data | Joint Evaluation | SR3D | NR3D | ScanRefer |
|---|---|---|---|---|
| Base+ScanNet++ | 0.632 | 0.674 | 0.562 | 0.608 |
| Base+ScanNet++ (updated task) | 0.631 | 0.668 | 0.568 | 0.612 |
| Base+ScanNet | 0.618 | 0.664 | 0.524 | 0.603 |

Table 10: Impact of scene diversity. We train all models on SR3D+NR3D+ScanRefer and add 30K samples from L3DD. We ablate whether these extra samples come from ScanNet or ScanNet++. We also ablate whether we are using the first or second iteration of the annotation instructions.

+ 30K L3DD samples from ScanNet++ (i.e., same quantity, but better quality). Table 10 shows that training on better quality scenes (2) outperforms (1) by about 1.5 percent. The use of data collected from our updated instructions of our task (high-distractor objects as targets, multi-anchor) does not significantly impact performance.

Table 11: **Robot Experiments**

| Model | Task Success |
|---|---|
| Concept Fusion | 2/10 |
| VLM (Llama-3) | 5/10 |
| VLM (GPT-4o) | 5.66/10 |
| LOCATE 3D+ (ours) | **8/10** |

# E. Real-World Robot Experiments

## E.1. Robot Experiment Details

LOCATE 3D is intended to work in the real world, the robot trials test the performance of the models in a real setting. These trials serve as a benchmark of evaluating different models. We are using Boston Dynamics Spot robot to carry out these experiments. The scene, a model apartment, this differs considerably from our training conditions by being multi-room test apartment and a much larger area.

We scan the environment using iPhone's Record3D app, which provides RGB-D data that is processed by our preprocessing pipeline into 5cm featurized voxel maps. The testing environment represents a challenging out-of-distribution scenario, featuring rooms regions and object categories unseen during training. The full scan of this apartment was processed offline to get detected bounding boxes for 10 pre-selected reference expression. Each of these references to a unique furniture and the prompts were 1.kitchen island, 2. rectangular dining table, 3. sink, 4. night stand near the window, 5. bed with pillow, 6. dresser in the office, 7. dresser in the hallway, 8. coffee table in front of the sofa, 9. desk in office, 10. kitchen counter with stove

We evaluate against two baselines: (1) A VLM-based pipeline using GPT-4/Llama and SAM2, and (2) ConceptFusion, a 3D mapping system. Results in Table 11 show LOCATE 3D+ achieves 8/10 performance.

The trial had tasks like "navigate to the coffee table in from of the sofa and pick up the plush toy". A given episode had 3 parts: location, navigation then pick up. Out of these only localisation was expected to be done via the model being benchmarked. Our navigation method uses a heuristic that combines A* path planning with point cloud data to identify a point near to the object. Specifically, this involves generating a low-resolution 2D occupancy grid from the point cloud, which enables the determination of accessible free spaces. The pick-up part of the task has a Boston Dynamics API call to grasp and OWL-ViT (Minderer et al., 2022) object detector. After completing navigation a *gaze policy* inspired by (Yokoyama et al., 2024) is used to scan the scene and the given object e.g "plush toy" was detected with object detector and the grasping API call was initiated. A closed set of objects for which it was tested out that the robot had high successes of grasping was used.

To ensure reliability, each task was repeated three times, resulting in a total of 30 trials per model on the Spot robot. A successful trial was defined as a sequence of three consecutive tasks: accurate localization, followed by effective navigation,

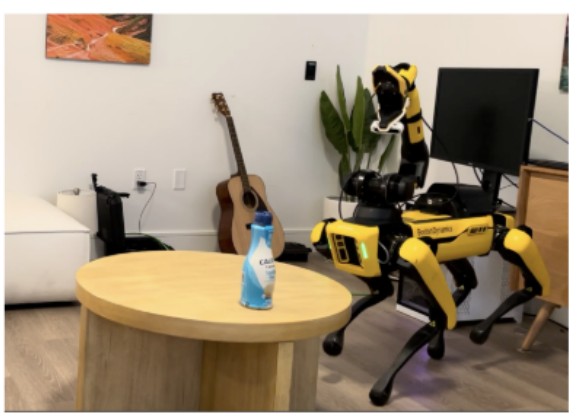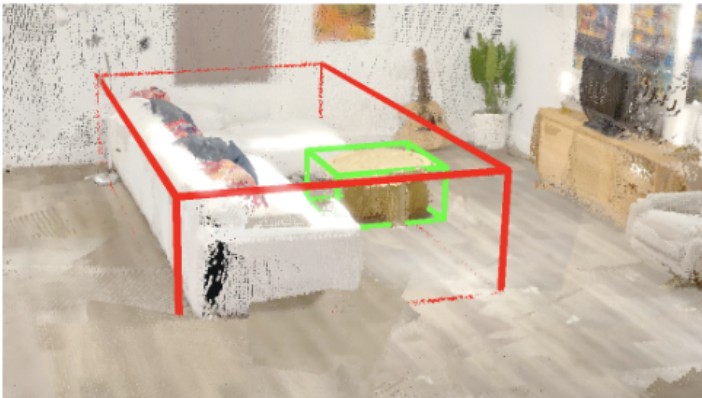

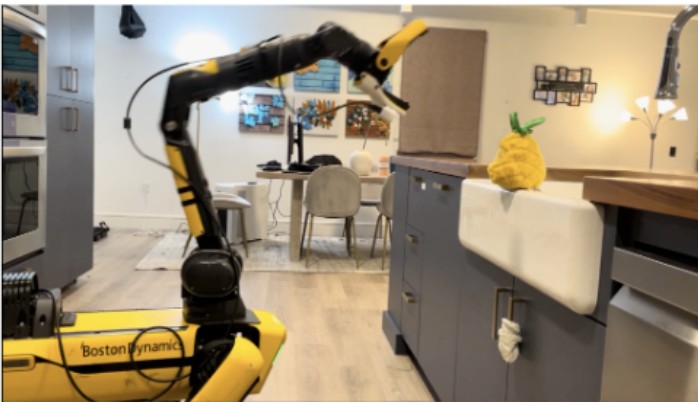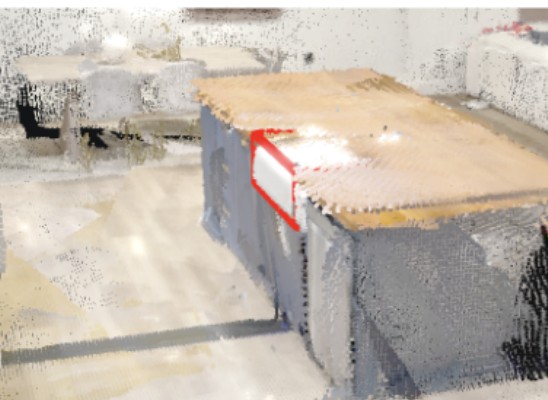

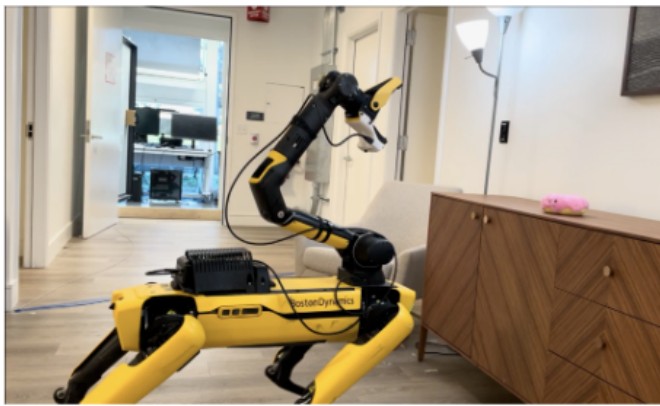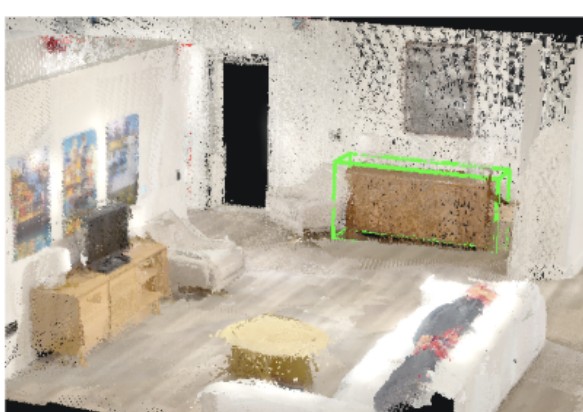

Figure 12: Examples of the Spot robot at the end of navigation task before the pick task (right) the output bounding boxes of LOCATE 3D+ model (left). The task (top) "navigate to *coffee table in front of the sofa* and pick up the bottle" (middle) "navigate to *sink* and pick up the pineapple plush toy" (bottom) "navigate to *the dresser in the hallway* and pick up pink plush toy"

and culminating in a successful pick action. Each trial had a binary outcome, with success contingent on the correct execution of all three tasks. We report the number of successful episodes averaged over the 3 repetitions achieved by each model. Notably, our results show that correct localization consistently facilitated successful navigation and pick-up.

An interesting finding is that coarse localization (within 2m of target) often proved sufficient given the robust manipulation capabilities of the pipeline. While triplicating results the VLM had a different seed value while generating the localization bounding boxes. The VLM pipeline being non-deterministic this gave different results. A successful detection for a particular seed value might not be replicated for all 3 tries. That is the reason why with averaging of these values a non-whole number

is part of the reported results.

We also observed that at times the most obvious way to refer to objects can include the room they are located in (e.g., "dresser in the bedroom"). However, our training data lacked such language, highlighting a need for future work to incorporate this type of spatial context.

## F. Vision-Language Model (VLM) Baselines

This section describes two baseline models that leverage vision-language models (VLMs) – Llama-3 (Meta AI, 2024) and GPT-4o (OpenAI, 2024) – within a modular pipeline designed to solve the 3D referring expressions task, which is similar to (Xu et al., 2024). Specifically, both baselines include three primary modules: (1) frame selection, (2) 2D object selection, and (3) 2D-to-3D lifting. Results for these VLM baselines are in Tables 1 and 4 and method details are provided below.

**Frame Selection.** In our Llama-3 (Meta AI, 2024) VLM baseline, frame selection is performed by captioning ($K = 20$) uniformly sampled RGB frames from the observation stream. Specifically, for each frame we use SAM-2 (Ravi et al., 2024) to segment the image, and set-of-mark prompting (Yang et al., 2023b) with Llama-3 to generate a list of objects in the frame, which we treat as the frame caption. Next, we past the $K = 20$ captions to Llama-3 along with the referring expressions query, and ask the model to select the top-3 frame that may include the object described in the query. Finally, for each of the top-3 frames, we ask Llama-3 to verify if object described in the referring expressions query is visible. The first verified frame is used in the subsequent modules. If none of the top-3 frames are verified, one is selected at random.

In our GPT-4o (OpenAI, 2024) VLM baseline, frame selection is performed by passing the $K = 20$ uniformly sampled RGB frames along with the referring expressions to the VLM, and asking the model to select the frame the includes the object described in the query. The selected frame is subsequently passed to the next module for 2D object selection.

**2D Object Selection.** In both baselines, object selection is performed using an open-vocabulary object detector (GroundingDINO (Liu et al., 2023)), a promptable image segmentation model (SAM 2 (Ravi et al., 2024)), and a VLM (Llama-3 (Meta AI, 2024) or GPT-4o (OpenAI, 2024)). Specifically, the full user query is passed to the object detector to generate 2D bounding boxes for candidate objects in the scene. Instance segmentation masks for each object are generated by SAM 2 and used to create a set-of-marks prompt (Yang et al., 2023b) for the VLM, which selects the object and correspondingly a 2D object mask.

**2D-to-3D Lifting** In the final module, the 2D object mask is "lifted" (or unprojected) into 3D. Specifically, we first use SAM 2 (Ravi et al., 2024) to propagate the 2D object mask over time. Next, we filter the 2D propagated mask, then unproject them into a 3D object point cloud using the depth observations and camera information (pose and intrinsics). Finally, we use DBSCAN to find the largest cluster in 3D object point cloud, and use the bounds as the predicted bounding box.

## G. Full Ablation table

Table 12 provides a complete set of results for all of the LOCATE 3D and LOCATE 3D+ experiments and ablations in this work. For all methods, we provide results on SR3D, NR3D, ScanRefer (Achlioptas et al., 2020; Chen et al., 2020). For most methods, we additionally provide results for the ScanNet++ (Yeshwanth et al., 2023) and ARKitScenes (Dehghan et al., 2021) splits from LOCATE 3D DATASET.

Table 12: **Comparison of various ablations and baselines.** Metrics shown as (*@25, @50*) for SN Joint, SR3D, NR3D, ScanRefer, SN++, and ARKitScenes.

| Input Features | Encoder Init | Training Data | Other | SN Joint | | SR3D | | NR3D | | ScanRefer | | SN++ | | ARKitScenes | |
|---|---|---|---|---|---|---|---|---|---|---|---|---|---|---|---|
| | | | | @25 | @50 | @25 | @50 | @25 | @50 | @25 | @50 | @25 | @50 | @25 | @50 |
| **Ours** | | | | | | | | | | | | | | | |
| (LOCATE 3D) CLIP-L + DINOv2 + SAM-H | 3D-Jepa | SN | - | 0.617 | 0.494 | 0.658 | 0.529 | 0.537 | 0.405 | 0.599 | 0.496 | 0.567 | 0.282 | 0.462 | 0.174 |
| (LOCATE 3D+) CLIP-L + DINOv2 + SAM-H | 3D-Jepa | SN + LX3D | - | 0.637 | 0.513 | 0.682 | 0.548 | 0.561 | 0.432 | 0.611 | 0.509 | 0.839 | 0.575 | 0.576 | 0.208 |
| **Input Featurization Ablations** | | | | | | | | | | | | | | | |
| CLIP-B + MobileSAM | 3D-Jepa | SN | – | 0.537 | 0.394 | 0.566 | 0.417 | 0.475 | 0.328 | 0.529 | 0.403 | – | – | – | – |
| Clip-L + SAM-H | 3D-Jepa | SN | – | 0.592 | 0.455 | 0.627 | 0.477 | 0.521 | 0.383 | 0.582 | 0.469 | – | – | – | – |
| DinoV2 | 3D-Jepa | SN | – | 0.537 | 0.421 | 0.560 | 0.437 | 0.468 | 0.352 | 0.546 | 0.445 | – | – | – | – |
| **Random Initialization Ablations** | | | | | | | | | | | | | | | |
| CLIP-L + DINOv2 + SAM-H | Random | SN | – | 0.588 | 0.463 | 0.628 | 0.496 | 0.516 | 0.382 | 0.567 | 0.461 | 0.515 | 0.255 | 0.417 | 0.152 |
| CLIP-L + DINOv2 + SAM-H | Random | SN + LX3D | – | 0.607 | 0.475 | 0.644 | 0.506 | 0.544 | 0.399 | 0.586 | 0.473 | 0.746 | 0.476 | 0.556 | 0.240 |
| **RGB Input Ablations** | | | | | | | | | | | | | | | |
| RGB | Random | SN | – | 0.513 | 0.386 | 0.530 | 0.400 | 0.449 | 0.321 | 0.530 | 0.410 | 0.375 | 0.150 | 0.113 | 0.042 |
| RGB | Random | SN + LX3D | – | 0.562 | 0.445 | 0.583 | 0.457 | 0.493 | 0.377 | 0.574 | 0.474 | 0.740 | 0.497 | 0.178 | 0.064 |
| **"Encoder-Free" Ablations** | | | | | | | | | | | | | | | |
| CLIP-L + DINOv2 + SAM-H | N/A | SN | – | 0.538 | 0.397 | 0.562 | 0.414 | 0.463 | 0.328 | 0.551 | 0.418 | 0.461 | 0.179 | 0.218 | 0.058 |
| RGB | N/A | SN | – | 0.289 | 0.174 | 0.278 | 0.174 | 0.247 | 0.133 | 0.342 | 0.205 | 0.158 | 0.046 | 0.009 | 0.000 |
| **Supervision and Decoder Heads Ablations** | | | | | | | | | | | | | | | |
| CLIP-L + DINOv2 + SAM-H | 3D-Jepa | SN | Naive Bbox | 0.554 | 0.391 | 0.598 | 0.426 | 0.464 | 0.304 | 0.542 | 0.391 | 0.545 | 0.260 | 0.458 | 0.177 |
| CLIP-L + DINOv2 + SAM-H | 3D-Jepa | SN | DBSCAN Bbox | 0.584 | 0.416 | 0.633 | 0.455 | 0.487 | 0.330 | 0.565 | 0.408 | 0.577 | 0.282 | 0.524 | 0.222 |
| CLIP-L + DINOv2 + SAM-H | 3D-Jepa | SN | MLP Bbox | 0.356 | 0.135 | 0.385 | 0.149 | 0.286 | 0.102 | 0.356 | 0.133 | 0.032 | 0.002 | 0.017 | 0.001 |
| CLIP-L + DINOv2 + SAM-H | 3D-Jepa | SN | No Mask (our head) | 0.003 | 0.000 | 0.003 | 0.000 | 0.002 | 0.000 | 0.005 | 0.000 | 0.001 | 0.000 | 0.000 | 0.000 |
| CLIP-L + DINOv2 + SAM-H | 3D-Jepa | SN | No Mask (mlp head) | 0.294 | 0.100 | 0.311 | 0.102 | 0.239 | 0.078 | 0.306 | 0.115 | 0.025 | 0.003 | 0.011 | 0.000 |
| **Decoder Scaling Ablations** | | | | | | | | | | | | | | | |
| CLIP-L + DINOv2 + SAM-H | 3D-Jepa | SN | Small Decoder | 0.581 | 0.433 | 0.623 | 0.465 | 0.496 | 0.348 | 0.567 | 0.439 | 0.494 | 0.206 | 0.356 | 0.106 |
| CLIP-L + DINOv2 + SAM-H | Random | SN | Small Decoder | 0.544 | 0.390 | 0.584 | 0.415 | 0.460 | 0.314 | 0.532 | 0.400 | 0.448 | 0.190 | 0.350 | 0.121 |
| CLIP-L + DINOv2 + SAM-H | 3D-Jepa | SN | Base Decoder | 0.606 | 0.476 | 0.652 | 0.511 | 0.524 | 0.389 | 0.583 | 0.476 | 0.552 | 0.267 | 0.450 | 0.165 |
| CLIP-L + DINOv2 + SAM-H | Random | SN | Base Decoder | 0.576 | 0.445 | 0.615 | 0.478 | 0.505 | 0.367 | 0.557 | 0.442 | 0.495 | 0.231 | 0.395 | 0.149 |
| **Training Ablations (Finetuning and Scheduler)** | | | | | | | | | | | | | | | |
| CLIP-L + DINOv2 + SAM-H | 3D-Jepa | SN | Frozen | 0.562 | 0.440 | 0.588 | 0.460 | 0.503 | 0.374 | 0.557 | 0.453 | 0.486 | 0.230 | 0.372 | 0.124 |
| CLIP-L + DINOv2 + SAM-H | 3D-Jepa | SN | Naive Scheduler | 0.595 | 0.470 | 0.638 | 0.503 | 0.512 | 0.383 | 0.578 | 0.475 | 0.524 | 0.248 | 0.498 | 0.203 |
| CLIP-L + DINOv2 + SAM-H | Random | SN | Our scheduler | 0.598 | 0.470 | 0.642 | 0.502 | 0.524 | 0.393 | 0.571 | 0.468 | 0.532 | 0.265 | 0.439 | 0.168 |
| **Other Baselines** | | | | | | | | | | | | | | | |
| RGB | PonderV2 | SN | – | 0.432 | 0.309 | 0.431 | 0.312 | 0.386 | 0.262 | 0.469 | 0.340 | 0.297 | 0.120 | 0.048 | 0.012 |

