# OpenReview forum: "LOCATE 3D: Real-World Object Localization via Self-Supervised Learning in 3D"
_ICML.cc/2025/Conference — ICML 2025 spotlightposter_

### Official Review · Reviewer_gKhM · 2025-02-28

**Overall Recommendation:** 4

**Summary:**

This paper proposes a method called Locate3D, which addresses the task of referring expression-based localization in 3D. The method consists of three key stages: (1) pre-processing stage in which an input RGB-D stream is processed using 2D foundation models SAM, CLIP and  DINOv2 in order to obtain 2D features which are then lifted to 3D following the ConceptFusion framework. The resulting point cloud representation associates each point with RGB, CLIP and DINOv2 features. (2) This representation is then used to first train a transformer-based encoder following a self-supervised learning framework called 3D-JEPA. This encoder is trained by using the self-supervised task of learning to predict per-point features in regions that are masked-out by a region sampling process. (3) Next, a decoder that takes as input the embedding representation obtained from the 3D-JEPA encoder, as well as a referring-expression is trained using the tasks of dense-3D segmentation as well as 3D bounding box detection. The model is trained using existing referring expression-based localization datasets as well as the LX3D dataset, which is a dataset proposed in this submission with the aim of extending existing 3D training datasets (ScanNet, ScanNet++, ARKitScenes) with referring-expression annotations in an automatized manner to obtain a referring expression dataset with a larger number and variety of scenes.

### Update after rebuttal
The clarifications and discussions provided in the rebuttal have sufficiently addressed my concerns. I am still leaning towards acceptance, and will be keeping my original score (4: Accept).

**Claims And Evidence:**

1. _[L011-L014, right column]_ The claim "They (existing 3D RefExp models trained on small benchmark datasets) often require human annotation at inference time in the form of detailed 3D meshes or object instance segmentation, making them difficult to deploy on real-world devices" is not fully accurate in my opinion. Such methods can use human annotated object masks for instance, but currently there are many strong 3D object instance mask predictors, 3D object detection methods and improved mesh reconstruction methods. The fact these methods were evaluated using GT boxes etc. does not necessarily make them unsuitable to real-world device deployment. I think this is important to clarify as this is critical for positioning the paper.
2. _[Title]_ "Locate3D: Real-World Object Localization via Self-Supervised Learning in 3D": This is not too critical and a relatively minor point: At the first glance, the title leads the reader to start with the assumption that the whole training process takes place within a self-supervised learning framework, whereas the proposed model actually indeed uses GT annotation data consisting of target object localization data paired with the referring expression queries.
3. _[L029-L033]_  It appears to me that the claim "Locate3D operates directly on sensor observation streams without requiring manual post-processing (e.g., 3D mesh refinement or ground-truth instance segmentations), making it readily deployable on robots and AR devices" would mean that given an RGB-D sequence, all processing and inference can take place in close-to real-time. However, I was unable to see an analysis or data on the runtime of the method. If my understanding is correct, the 2D feature computation using SAM/CLIP/DINO also takes place during inference time, and I would expect this part to take a few minutes. This could mean that the claims about real-world deployment readiness might need to be revised.
4. _[L086-089]_  Similarly, I find the claim "Locate3D does not require ground-truth region proposals, meshes, or surface normals, making it suitable for real-world deployment." partially incorrect if I have a correct understanding of the method. It appears to me that the method Locate3D indeed requires ground-truth region proposals, but only during training. But the claim I quoted sounds like the method never needs such paired annotations of segmentation and referring expressions. Please correct me if I misinterpret this statement or perhaps misunderstand this aspect of the proposed method.

**Essential References Not Discussed:**

Related works are generally well discussed, but there are a few works that could be added. For instance, ConcreteNet (Unal et al, ECCV 2024, _"Four Ways to Improve Verbo-visual Fusion for Dense 3D Visual Grounding"_) only takes a point cloud as input, which appear to be under very similar assumptions as this work. In L394-395 of the submission, it is mentioned that "few prior works operate under this setting", but according to my understanding ConcreteNet follows this setting as well.

**Experimental Designs Or Analyses:**

I found that the experimental design is well-founded in general. The paper follows the established benchmarks and evaluation methodologies for 3D-RefExp task. There are plenty of ablation studies analysing different aspects of the method: input features, encoder configurations and the use of different foundation models. Additionally, I also appreciated experiments on cross-dataset evaluation as well as real-world deployment with a mobile robot setup.

**Methods And Evaluation Criteria:**

Proposed method is reasonable, successfully targeting the 3D referring expression-based localization task. Proposed method and its components are thoroughly ablated. Evaluation criteria, metrics and chosen benchmark datasets follow the commonly employed evaluation methodology for the task. There are also additional results on the proposed LX3D dataset.

**Other Comments Or Suggestions:**

*Minor Comments*
- 3D-JEPA term and what it stands for is never explicitly explained, at least not in the main paper. While there is a small reference to the JEPA framework (Assran et al.) in L135, this is still very unclear, as the broader audience/3D scene-understanding community might not be familiar with this particular framework.

*Typos and grammatical issues:*
- _L094 (left column) "Finally, it also successfully localised objects ...":_ "localises" might be better suited to the flow.
- _L117 (left column) "The point cloud_ $\mathrm{PtC_{lift}}$ _can be directly use for object localization ...":_ should be "used".
- L676 (appendix) "... figure 5 ...":_ should be "Figure 5".
- _L1038 (appendix) "... while the PCA projections reveals that the encoder ...":_ should be "reveal".
- _L1039 (appendix) "... while the decoder learns more sharped localized features ...":_ the term "sharper" might be better instead of "more sharped".

**Other Strengths And Weaknesses:**

*Strengths*
- The paper addresses a relevant and important task (3D referring-expressiong based localization). I also appreciated that the proposed method aims to target more realistic conditions with sensor PC instead of refined meshes.
- The paper is generally written well with clarity.
- I found that the proposed method is reasonable, and it is not an overly-complicated framework.
- A very systematic and detailed evaluation of the method components is presented.
- There are additional data contributions (if they will be publicly released)

*Weaknesses*
- I found that the level of detail regarding the 3D-JEPA framework could be improved as currently it reads a bit high-level. However, as this is one of the main proposed contributions of the paper, I would have expected to see a much detailed discussion on 3D-JEPA, as well as how it relates to the original JEPA framework, especially considering that many readers in the 3D scene understanding domain might not necessarily be familiar with the JEPA framework.
- I did not find a thorough discussion on the limitations of the proposed method.

**Questions For Authors:**

- Is the LX3D dataset proposed in this work planned to be publicly released?
- I am a bit unclear about the inference setting. At inference time, does the method still require the RGB-D stream and does it still need to use SAM/CLIP/DINO to obtain initial per-point features? If so, I would expect the feature extraction to take a few minutes for a given small/medium sized indoor scene, which might affect the claims regarding how much the proposed method is suitable for real-world deployment. As one of the main claims is "Locate3D operates directly on sensor observation streams without requireing manual post-processing (e.g., 3D mesh refinement or ground-truth instance segmentations), making it readily deployable on robots and AR devices" ([L029-L033]), how much time it takes to get the predictions is critical.
- How is the generalizability of the method in terms of identifying novel classes affected by the contextualization of the features via the 3D-JEPA encoder? I understand that this process would enable the method to have less noisy features that are also more informative about the spatial relationships between scene objects. However, at the same time would this not result in a decline in novel-object localization performance? Also in the context of the 3D-RefExp task, I would expect the localization performance w.r.t. object-attribute based queries (such as "red chair") might also be negatively affected as the scene features now cross-attend to each other. Is there any component that is designed to addresses the preservation of object-attribute related features?
- Generally, it would be great if a short discussion on the points listed in the "Claims and Evidence" section can be provided, especially number 3 and 4 (which mainly discuss the same aspect)
- How is the mask and patch feature aggregation performed? I wasn't fully clear about this from L107.

**Relation To Broader Scientific Literature:**

3D-JEPA: This framework very closely follows the original JEPA framework in ideation. I also have identified a few other extensions of the JEPA framework to 3D. Generally I find that the idea is not necessarily novel, but designing 3D-JEPA in the context of improving per-point features lifted from VLM models is a novel idea as far as I can judge. LX3D dataset is generally similar to existing benchmarks for 3D-RefExp task, but its key benefit appears to be the size of the dataset and the automatized generation of referring expression captions. However, such automatic caption generation methods were indeed proposed in prior work. The claims about the proposed contribution about achieving strong 3D-RefExp results "with fewer assumptions compared to prior models" need to be made a bit more sound for us to be able to evaluate the novelty of Locate3D a bit better.

**Theoretical Claims:**

N/A

---

> ### Author Rebuttal · Authors · 2025-04-01
>
> We appreciate the reviewer for the thorough reading of our manuscript and valuable comments.  Below we address each point raised in detail
>
> **Clarification regarding our LX3D**
>
> We noted the reviewer mentions our dataset being automatically generated, we'd like to clarify that our dataset is human-collected. Annotators curated and validated every sample. We invite the reviewer to read more details about our annotation setup in Appendix D.
>
> **Runtime analysis and assumptions [L029-L033]**
>
> Indeed, extracting 2D features and lifting them to 3D is computationally expensive. We address this by caching the environment’s 2D features for each view, as well as the featurized point cloud. For ScanNet experiments, we compute this cache offline; and for robot experiments we compute it while doing the initial environment exploration phase. With this feature cache, a forward pass of our model takes ~1 second for a scene with ~100k feature points and utilizes ~8 GB of VRAM on an A100 GPU.
>
> We can utilize such caching because our benchmarks operate under static (ScanNet) or quasi-static (robot) environments. Extending our approach to dynamic scenes would require real-time 2D feature computation and continuous updates to the featurized environment. We believe the former is a matter of engineering, while the latter is an active research area, explored by methods like Lifelong LERF (A. Rashid, 2024). We will include these assumptions in our limitations section.
>
> **[L086-089] rewording**
>
> We propose rewording this claim to more accurately state: "Locate 3D does not require ground-truth region proposals, high-quality mesh sampled pointclouds, or surface normals at inference time, making it suitable for real-world deployment."
>
> **Clarification on positioning**
>
> We propose revising [L011-L014, right col] to clarify: "Many existing 3D RefExp models are evaluated on benchmark datasets with pointclouds sampled from high-quality post-processed 3D meshes or object instance segmentations already available. In contrast, Locate 3D operates directly on raw sensor observation streams without requiring intermediate processing steps like mesh reconstruction, instance segmentation, or pre-trained object detectors at inference time."
>
> We want to highlight that most previous work has been trained and evaluated utilizing mesh pointclouds, while our approach leverages pointclouds obtained directly from RGB-D sensors (please see our response to reviewer C37t on this topic).
>
> **Discussion of ConcreteNet**
>
> We will add ConcreteNet to the related works section and baselines in Table 1. We note that this method operates on ScanNet's mesh-sampled pointclouds rather than sensor pointclouds. While ConcreteNet achieves 56.12% Acc@25 on ScanRefer, our approach demonstrates superior performance (59.9%) under the more challenging setting.
>
> **3D-JEPA exposition**
>
> We appreciate the observation. In our revision, we will provide a clearer introduction to this concept in section 2.2 explaining (1) The core principles of the JEPA framework (2) Why adapting this approach to 3D pointclouds (3) Main deltas to the original framework.
>
> **Discussion on Limitations**
>
> We will add a limitations section discussing (1) our current approach's constraints with dynamic environments (2) our method being limited to few-room environments (potentially addressed by exploring RoPE positional embeddings or similar techniques).
>
> **Typos**
>
> We appreciate the careful proofreading, we will correct all of these in the revised manuscript.
>
> **Release of LX3D dataset**
>
> We will publicly release the LX3D dataset, trained Locate 3D model, and 3D-JEPA backbone upon publication.
>
> **Generalizability (novel-object localization)**
>
> We believe both questions can be boiled down to how well CLIP features are preserved through 3D-JEPA pre-training. Our linear probing experiments in Section 4.2 address this, particularly on the "noun correctness" task 3D-JEPA achieves 73% accuracy compared to ConceptFusion's 66%. These results demonstrate that 3D-JEPA not only preserves but enhances the semantic understanding from foundation models. This pattern mirrors developments in language modeling, where contextualized embeddings (like BERT) consistently outperform non-contextualized word embeddings for both common and rare words.
>
> **Mask and patch feature aggregation**
>
> For 3D points observed in multiple views, we aggregate features by weighted averaging. We voxelize the pointcloud (5 cm voxel size) and compute a single feature per voxel by weighted averaging all contained features. Weights are calculated using trilinear interpolation based on distance to voxel boundaries. This weighted averaging approach is common practice in both our baselines (e.g., ConceptFusion) and traditional RGB-D mapping literature (Real-time 3D Reconstruction in Dynamic Scenes using Point-based Fusion - ICCV 2013). We explored alternative aggregation methods (max pooling, simple average, summation) before selecting weighted averaging.

---

> > ### Comment · Reviewer_gKhM · 2025-04-02
> >
> > I would like to thank the authors for the rebuttal. The clarifications and discussions provided in the rebuttal have sufficiently addressed my concerns. I am still leaning towards acceptance, and will be keeping my original score (4: Accept).

---

### Official Review · Reviewer_fbHG · 2025-03-06

**Overall Recommendation:** 3

**Summary:**

This paper focuses on object localization via referring expressions in 3D real-world scenes to understand the 3D physical world, which is a valuable task in 3D perception and understanding. It introduces an end-to-end 3D transformer network to get point cloud, 3D features, and text query as input, and output the potential masks, box, and predicted text tokens. To get better 3D feature representation, it uses a novel self-supervised learning algorithm for 3D point clouds, 3D-JEPA, which is inspired by the 2D JEPA framework. The motivation is clearly explained in the paper.
1. A novel SSL method for 3D point clouds, 3D-JEPA. It demonstrates that by adopting the pre-trained features from 3D-JEPA, the global perception capability is greatly enhanced. It can also provide improvements for downstream locate3D tasks in both in-domain and out-of-domain scenarios.
2. An end-to-end 3D transformer model for the 3D referring expression task, locate3D. It leverages the overall performance on public 3D benchmarks like ScanRefer, SR3D, and NR3D.
3. An incremental 3D referring expression dataset, LX3D. The samples are originally captured in ScanNet, ScanNet++, and ARKitScenes, and expand the language annotations.

**Claims And Evidence:**

none

**Essential References Not Discussed:**

1. 3D-JEPA: A Joint Embedding Predictive Architecture for 3D Self-Supervised Representation Learning

**Experimental Designs Or Analyses:**

In Table 1:
E1. The comparison with GPT-related methods seems unfair. How is the VLM+baseline method implemented is not discussed in the paper. Did it utilize depth information?
E2. How does it compare with other 3D-LLM methods, such as 3D-LLM[1], Leo[2], or related works, which have also been implemented on datasets like ScanRefer?
E3. There is a lack of comparison with other 3D semantic alignment and scene-level features.

In Table 2:
E4. Is PTV3 trained from scratch or using pretrained weights?
E5. Why does random initialization perform better than the 3D-JEPA fixed scenario? What is the context of the 3D-JEPA Frozen example, and please provide an explanation.

[1] 3D-LLM: Injecting the 3D World into Large Language Models
[2] An Embodied Generalist Agent in 3D World

**Methods And Evaluation Criteria:**

M1. The idea of deriving btter 3D features from 2D pre-trained features seems somewhat unconvincing. How do you demonstrate that the 3D capability is obtained based on 2D pre-trained features. Although PTV3 is used as the encoder, the approach is still primarily based on 2D features and their spatial contextual proximity for supervision. Therefore, I have some concerns about the name "3D-JEPA".

M2. What is the strategy for aggregating multi-view features for the same point? Does this strategy affect the accuracy of the final referring expression task? How do you handle multiple features from different perspectives for the same point?

M3. During the 3D-JPEA training, to what extent can the mask be achieved? Have you considered mask the points with non-random yet reasonable sampling methods such as farthest point sampling ?

**Other Comments Or Suggestions:**

Please see above.

**Other Strengths And Weaknesses:**

None.

**Questions For Authors:**

Q1. I am very interested in understanding what additional information, at the feature level, is provided by 3D-JEPA training compared to the original 2D pre-trained features. Additionally, how can this be demonstrated experimentally?
Q2. I am a little confused the metrics and settings in table1 when it compares to other GP4-4o methods. For those MLLM  methods, how to locate the 3d bbox?

**Relation To Broader Scientific Literature:**

None.

**Theoretical Claims:**

none

---

> ### Author Rebuttal · Authors · 2025-04-01
>
> We thank the reviewer for their detailed feedback. Their comments have helped us identify areas where we can better explain and justify our technical contributions. We address each comment in detail below.
>
> **Deriving better 3D features from 2D features; 3D-JEPA**
>
> The 3D nature of our approach comes from the architecture and learning objective rather than the individual pointwise inputs. Our model processes the entire featurized pointcloud using explicit 3D coordinates to produce per-point features. 3D-JEPA's objective forces the model to learn additional information beyond 2D features through masked prediction in 3D space, requiring it to infer features for occluded regions using contextual information from the surrounding visible 3D scene. In short, the "3D" in 3D-JEPA refers to the domain in which the representation learning occurs.
>
> Our approach has precedent in prior work like OpenScene, which demonstrated that distilling 2D vision-language features to 3D via contrastive learning results in 3D models that outperform the original 2D features.
>
> Empirical evidence shows 3D-JEPA features encode valuable information beyond lifted 2D features:
> 1. 3D-JEPA features demonstrate superior performance in linear probing experiments (Section 4.2) when compared to ConceptFusion features (34%→39% and 66%→73%)
> 2. Frozen 3D-JEPA features provide ~5% improvement over ConceptFusion inputs to the decoder (Table 2)
> 3. ConceptFusion (direct 2D-to-3D feature lifting) achieves 20% success on our robot evaluations compared to Locate 3D's 80% (Table 9)
>
> **Aggregating multi-view features**
>
> We voxelize the pointcloud at 5 cm resolution and aggregate multi-view point features within each voxel using weighted averaging based on trilinear interpolation distances. This aligns with standard practice in prior works (e.g., ConceptFusion); given space constraints, please also refer to our response to reviewer gKhM.
>
> **Farthest point sampling for masking**
>
> Yes, we tried farthest point sampling but did not see improvements. We additionally experimented with several other masking strategies as discussed in Appendix A.1. However, none of the methods improved over random masking (similar to the findings in Assran et. al., 2023). Thus, we use random masks for their computational efficiency.
>
> **Fairness of VLM baselines and usage of depth information**
>
> Yes, the two VLM baselines use depth information. In fact, they use the exact same inputs as Locate 3D (RGB-D + camera pose and intrinsics) to ensure fair comparisons. The VLM baselines are detailed in Appendix F.
>
> At a high level, these baselines use a modular pipeline that consists of first selecting an RGB frame using a VLM, then selecting an object in the 2D frame using GroundingDINO, SAM-2, and a VLM, and finally determining a 3D bounding box using depth and camera extrinsic and intrinsic information.
>
> **Comparison with 3D-LLM and LEO on ScanRefer**
>
> Locate 3D significantly outperforms 3D-LLM [1]. Specifically, 3D-LLM reports 30.3% Acc@0.25 on ScanRefer, while Locate 3D achieves 59.9% (Table 1). We will add results from 3D-LLM [1] to Table 1. LEO [2] is evaluated on ScanQA but not on ScanRefer (note that ScanQA assumes text outputs, while ScanRefer requires bounding boxes or instance masks). We will discuss LEO in related work.
>
> **Comparison with other 3D semantic alignment and scene-level features**
>
> Please refer to our previous response for a comparison of our approach to 3D-LLM, which trains a model to consume the entire scene. We are also happy to discuss additional related work if you might have concrete suggestions.
>
> **PTV3: Scratch vs pre-trained in Table 2**
>
> In our approach, PTV3 is pre-trained with 3D-JEPA. In Table 2, we study alternatives including “random” initialization (i.e., PTV3  is initialized from scratch), and initializing from “PonderV2” weights. In all three cases, the encoder is finetuned end-to-end for referential grounding. We will add these details to the Table 2 caption.
>
> **Table 2: Random initialization vs. 3D-JEPA (Frozen)**
>
> The better performance of random initialization compared to frozen 3D-JEPA can be attributed to the significant number of extra trainable parameters (~250M) that are optimized specifically for the referential grounding task. When we freeze the 3D-JEPA encoder, we only train the decoder and significantly limit the model's ability to adapt the encoder parameters to the task at hand. However, it's important to note that fine-tuning the 3D-JEPA encoder (our full model) provides the best performance, showing that while trainable model capacity is important, the pre-trained weights offer valuable initialization that leads to better results when allowed to adapt to the target task.
>
> **Table 1 clarifications (metrics, setting, VLM)**
>
> The VLM baselines (LLaMA and GPT-4o) use the exact same metrics and settings as other methods (such as Locate 3D) in Table 1. Specifically, the VLM baselines return 3D bounding boxes as detailed in Appendix F.

---

> > ### Comment · Reviewer_fbHG · 2025-04-03
> >
> > Thanks for the rebuttal, the replies address my primary concerns effectively. So I raise my rating as weak accept.

---

### Official Review · Reviewer_C37t · 2025-03-12

**Overall Recommendation:** 4

**Summary:**

The paper proposes Locate 3D, a model for 3D grounding which achieves SOTA performance and strong out-of-domain generalization.
Specifically, a novel self-supervised learning method, 3D-JEPA, is proposed, generating contextualized features for the scene. Also, a new dataset LX3D is proposed to test the robustness of the proposed model.

## update after rebuttal
Please see the rebuttal comment below.

**Claims And Evidence:**

Yes.

**Essential References Not Discussed:**

No.

**Experimental Designs Or Analyses:**

The proposed new dataset LX3D contains data from ScanNet and ScanNet++, where the visual data are similar. This poses some question in terms of the effectiveness on evaluating the robustness for out-of-domain data.

**Methods And Evaluation Criteria:**

The proposed methods and evaluation are reasonable.

**Other Comments Or Suggestions:**

For table 1, please bold the best result for Acc@50 on ScanRefer for consistency.

**Other Strengths And Weaknesses:**

Strength:
1. The paper is well-written, with clear structure and illustrations.

Weakness:
1. It's unclear how the authors define mesh PC and sensor PC. The reviewer notice the section C.3 in the appendix, but still get confused about the categories in Table 1. The authors could further clarify the input format for different methods, and the key difference for this paper.

**Questions For Authors:**

1. It's unclear how the authors define mesh PC and sensor PC. For the mesh pc methods in Table 1, the reviewer believe most of them are using point cloud as input, not mesh.
2. For 3D-VisTA, how do you evaluating under Mesh PC and Sensor PC + Proposals from Mesh PC settings? Why not Sensor PC and Sensor PC + Proposals from Mesh PC settings, which is more comparable.
3. Based on Figure 3, how do you choose the final prediction from the Q generated box? In the related work section (line 404-right column), the authors mention that 3D grounding approaches that 'provide probabilities for region proposals' is 'often difficult in 3D, and are prone to failures'. The reviewer is curious about the differences.
4. The proposed new dataset LX3D contains data from ScanNet and ScanNet++, where the visual data are similar. This poses some question in terms of the effectiveness on evaluating the robustness for out-of-domain data. Also from the results in table 4, the LOCATE 3D is not the best in multiple subsets.
5. In table 1, it is interesting that the proposed methods outperforms the existing methods in Acc@25, but not on Acc@50. Do the authors have some intuition why it is the case?
6. In table 3, it is unclear why the authors choose to use different visual backbones. Does CLIP+DINO indicates using different backbones for masks and image separately, or using both features for both masks and the image? If it is the former one, why using different backbones for masks and image? Why not directly using CLIP for both or DINO for both?

The reviewer may adjust the final rating after rebuttal based on the clarifications from the authors.

**Relation To Broader Scientific Literature:**

This paper proposes a framework, which enables 3D grounding on posed RGB-D frames. It shows the real-world deployment on robots, and could potentially extend to AR devices.

**Theoretical Claims:**

No theoretical claims and proofs involved.

---

> ### Author Rebuttal · Authors · 2025-04-01
>
> We thank the reviewer for their thorough feedback. Below we address each point in detail
>
> **Clarification on mesh PC vs sensor PC**
>
> By "Mesh PC" we refer to pointclouds sampled from carefully reconstructed 3D meshes that undergo extensive post-processing. Most prior works use this format as it’s ScanNet’s default.
>
> In contrast, "Sensor PC" refers to pointclouds obtained directly from RGB-D sensors. These contain raw depth measurements including noise, missing regions, and registration errors. This format better represents real-world deployment scenarios.
>
> Methods originally reported on Mesh PC suffer significant performance drops (8-10%) when trained and evaluated on sensor data (Table 1), 3D-VisTA drops from 53.1% to 45.9% Acc@25 and BUTD-DETR from 50.28% to 40.7%. This gap was also observed by Odin (Jain et al, 2024). Under the (more rigorous) evaluation setting comprising sensor pointclouds, Locate 3D achieves SoTA (61.7% Acc@25) performance. We will clarify this terminology in the paper.
>
> > For the mesh pc methods in Table 1, the reviewer believes most of them are using pointcloud as input, not mesh.
>
> The reviewer is correct – while these methods use pointclouds as input, we use the term "Mesh pointcloud" to indicate that these pointclouds are sampled from cleaned, post-processed meshes.
>
> **3D-VisTA evaluation clarification**
>
> 3D-VisTA was originally designed for mesh pointclouds with a two-stage architecture: first using Mask3D to generate object proposals, then processing these proposals with the pointcloud for grounding. Due to implementation constraints and pre-computed proposal dependencies, we report results using sensor PC for 3D-VisTA while maintaining the original Mask3D proposals from mesh PC.
> We note that **this represents an upper bound for 3D-VisTA's performance** on sensor data, as using sensor PC for proposal generation would only decrease performance further. Even with this advantage, 3D-VisTA achieves 45.9% Acc@25, while Locate 3D significantly outperforms it at 61.7%.
>
> **Question about query selection**
>
> For ScanNet benchmarks we select the query with the highest “token probability” for the target noun. In our robot experiments, we use an off-the-shelf LLM to parse the referring expression and identify the target noun before applying the same selection mechanism.
>
> > Authors mention that 3D grounding approaches [...] are prone to failures'. The reviewer is curious about the differences.
>
> These approaches, such as 3D-VisTA (Zhu et al., 2023) and PQ3D (Zhu et al., 2024), rely on external object detectors to generate proposals without considering the language query and then train their model to select one. If the object proposal misses, the entire model is bound to fail. Single-stage approaches like ours and BUTD-DETR (Jain et al., 2022), directly predict the bounding box by jointly considering the scene and language tokens. We will incorporate this discussion into our paper.
>
> **LX3D out-of-domain evaluation clarification (ScanNet vs ScanNet++)**
>
> We see key differences in scanning hardware, setting (partial scene vs. full scene), scene size, and distribution of queries between the train and eval settings. Further, our eval set contains both ARKitScenes and ScanNet++. While we agree that some aspects of the visual distribution remain constant (e.g. common object classes), we believe that the evaluation faithfully represents the domain gap present during a new deployment of our model to indoor scenes.
>
> As for your latter point (Locate 3D not being the best in multiple subsets) – we believe the VLM approach is strongest on ScanNet++ because each input contains only 5 frames, and Locate 3D not achieving maximum performance on FRE is an artifact of the small eval set size (~250 annotations on 1 scene).
>
> **Better perf on Acc@25 but lower on Acc@50**
>
> This discrepancy arises because PQ3D operates on mesh pointclouds, while our method works on sensor pointclouds. At lower IoU thresholds (Acc@25), our approach effectively predicts the bounding box despite sensor data imperfections. However, at higher thresholds (Acc@50), achieving precise alignment becomes more challenging, resulting in a more significant performance drop for sensor-based methods. In short, PQ3D is evaluated in a more relaxed setting than ours, and is at an advantage.
>
> Despite our best efforts, we could not re-train PQ3D with sensor pointclouds due to their use of multiple backbones and multi-stage training strategies, making a direct comparison difficult.
>
> **CLIP and DINO backbones in Table 3**
>
> Whenever we use CLIP features, we first obtain masks using SAM and then extract CLIP embeddings for these masks following Conceptfusion. **This approach is necessary because CLIP produces global image embeddings, not patch or pixel-level features**. For DINO, we directly use its dense patch-level features and map them to individual points without requiring a segmentation step. We will add this explanation to Section 2.1.

---

> > ### Comment · Reviewer_C37t · 2025-04-02
> >
> > Thanks for the detailed rebuttal. The authors have addressed all my concerns.
> > I have also read the reviews from other reviewers and authors' rebuttal. I would like to increase my rating to accept.

---

### Official Review · Reviewer_J9xs · 2025-03-14

**Overall Recommendation:** 3

**Summary:**

This paper introduces Locate 3D, a model for localizing objects in 3D scenes from referring expressions, achieving state-of-the-art performance on standard referential grounding benchmarks. The key innovation is 3D-JEPA, a self-supervised learning (SSL) algorithm to learn contextualized scene representations. Locate 3D outperforms existing methods on SR3D, NR3D, and ScanRefer with fewer assumptions. Additionally, the LX3D dataset (130K+ language annotations) boosts generalization. Experiments demonstrate the model's strong performance and potential for deployment in real-world systems.

**Claims And Evidence:**

The claims made in the submission are generally well-supported by clear and convincing evidence. However, the claim about deployment could benefit from stronger evidence or additional clarification:
- “... making it readily deployable on robots and AR devices” is supported by the experiments in Sec. 4.4. While the paper demonstrates deployment on a robot, the AR application is not explicitly evaluated.

**Essential References Not Discussed:**

One of the core ideas of this work is to lift the general features produced by 2D foundation models. Therefore, I suggest discussing more about the existing work which conveys similar ideas in broader 3D scene understanding aspects. For example, OpenScene (CVPR’ 2023), Lift3D (CVPR’ 2024), etc..

**Experimental Designs Or Analyses:**

The experimental designs and analyses are well-structured and easy to follow. I have mentioned some points related to experiments in the previous sections to help strengthen this submission. In this part, it is also interesting to discuss how different 2D image features influence the performance of located 3D as it is mentioned in line 115-116 without experimental analysis.

**Methods And Evaluation Criteria:**

The proposed methods and evaluation criteria generally make sense for the 3D object localization task, but there are some areas for potential improvement.
- Locate 3D is designed for real-world deployment, but the current version of submission lacks the discussion of inference speed, memory usage, or computational cost.
- I understand LX3D improves performance, but it is still unclear whether the improvement is due to data volume or diversity?

**Other Comments Or Suggestions:**

Overall, I believe this paper presents a useful tool for the community, but the experiments and analysis are not sufficiently thorough, which weakens some aspects of its contribution. However, if the authors can address my concerns, I would consider adjusting my score accordingly.

**Other Strengths And Weaknesses:**

Locate 3D has the potential to be a useful tool for some important downstream applications regarding higher-level 3D understanding since localization could be fundamental operations for scene understanding.

Though the overall experimental structure in this submission is easy to follow, there are still some incomplete aspects regarding the LX3D dataset, especially considering that it is claimed as a core contribution. In this version of the submission, there is a lack of sufficient quantitative analysis (such as the aspects mentioned above) and qualitative analysis (e.g., annotation quality and granularity) of the LX3D dataset, which raises concerns about its reliability. Moreover, in the preprocessing stage, the authors mention using SAM to process images and obtain masks. However, SAM-generated masks do not ensure multi-view consistency, which is confusing to me. Using such masks for subsequent processing carries the risk of introducing excessive noise.

**Questions For Authors:**

I have the following key questions that, if addressed, could potentially change my evaluation of the paper:
1. Clarification on LX3D Dataset Analysis
2. Computational Efficiency and Scalability
3. Multi-View Consistency in SAM-based Preprocessing
4. Claimed AR related applications

For the main concerns listed above, details can be found in the previous sections. For other minor concerns mentioned, I believe addressing them could also help strengthen this submission.

**Relation To Broader Scientific Literature:**

The key contribution of the paper is closely related to some prior findings/results/ideas including (1) 3D object localization work, (2) Serf-supervised Learning (SSL) on 3D representations and (3) 2D foundation models. Different from previous 3D object localization work, Locate 3D operates directly on sensor observation streams, and works with fewer assumptions. 3D-JEPA is a SSL method inspired by previous SSL work like PointMAE, which is specially designed for 3D object localization in order to learn conceptualized representations of 3D scenes. Large-scale 2D foundation models like DINOv2 and CLIP are treated as tools by Locate 3D for a better capability of localization.

**Theoretical Claims:**

The paper primarily focuses on algorithmic and empirical contributions, rather than formal theoretical claims with proofs.

For line 113-115, authors claim the feature dimension “combines RGB, CLIP and DINOv2 features”. It is unclear how to combine those features as this is essentially an explanation of  $f_i$ in the corresponding equation.

---

> ### Author Rebuttal · Authors · 2025-04-01
>
> We thank the reviewer for their thorough review and constructive feedback. Below we address each point in detail.
>
> **Clarification about AR application**
>
> Indeed, we do not demonstrate deployment on an AR device. Our intended claim is that Locate 3D's input format is directly compatible with data streams from AR devices and robots - it operates on raw RGBD and camera parameters, which is the standard output format of these platforms. We demonstrate this compatibility through evaluation on diverse sensor-captured datasets (ScanNet, ScanNet++, ARKitScenes) as well as robot deployment. We will revise our phrasing to clarify that we're referring to input compatibility rather than explicit AR deployment.
>
> **Inference speed, memory usage, computational cost**
>
> We will add this discussion to the paper. Our inference pipeline operates in two phases. First, we compute and cache 2D features and a featurized pointcloud of the environment. This can be  done offline for static environments like ScanNet, or during an initial exploration phase for robots. With this cache in place, our model performs inference in approximately one second for a scene with ~100,000 feature points with an average peak VRAM of ~8.1 GB on one A100.
>
> **LX3D dataset analysis; data volume vs diversity**
>
> We ran an additional experiment, in which we find that **scene diversity is a key factor in LX3D improvements**. Specifically, we compare two conditions: (1) ScanNet training data + 30K LX3D samples also from ScanNet and (2) ScanNet data + 30K LX3D samples from ScanNet++ (i.e., same quantity, but better quality). We find that training on better quality scenes (2) outperforms (1) by ~2%. We will include this additional experiment in a revision.
>
> The annotation quality of every sample LX3D was validated by human annotators. Specifically, only samples marked as unambiguously correct are included in the final dataset to ensure reliability, more details about our annotation setup and dataset can be found in Appendix D.
>
> Furthermore, we will publicly release the LX3D dataset upon publication, providing a high-quality asset to support future research in 3D referential grounding.
>
> **Combining RGB, CLIP, DINO features**
>
> The features are combined by concatenating CLIP and DINO features with a harmonically embedded representation of the RGB features. Specifically, CLIP features are generated for SAM masks, with a learnable token used when mask features are unavailable. Full details are provided in Appendix A.2 and we will add a summary to the main paper. For more details, we invite the reviewer to read the response to a similar question posed by reviewer *gKhM*.
>
> **Impact of different 2D image features**
>
> We provide experimental analysis in Table 3. We observe that features extracted from larger models (CLIP-L, SAM-H) consistently outperform those extracted from models with fewer parameters (CLIP-B, MobileSAM), with CLIP-L achieving 59.2% Acc@25 compared to CLIP-B's 53.7%. Additionally, concatenating features from CLIP and DINOv2 significantly improves results compared to using either of these in isolation, reaching 61.7% Acc@25, indicating that each of the features provide complementary information that benefits 3D object localization.
>
> **Discussing related work that performs lifting of 2D features (OpenScene, Lift3D)**
>
> We appreciate the suggestion. Approaches such as OpenScene (CVPR 2023), Lift3D (CVPR 2024), and prior ideas proposed in distilled feature fields (Neurips 2022), similarly lift 2D features to 3D via neural rendering or contrastive learning. However, we incorporate two key advances in addition to lifting 2D features. (1) our self-supervised 3D pretraining (3D-JEPA) approach contextualizes the lifted 3D features, (2) our specialized language-conditioned decoder enables precise referential object localization. Both (1) and (2) contribute significantly to our performance. As shown in Table 9, Locate 3D+ (8/10) substantially outperforms Concept Fusion (2/10), a zero-shot feature lifting method, on complex referential tasks. We'll add this discussion to the related work section.
>
> **Multi-view inconsistency of SAM masks**
>
> Correct, SAM masks are not multi-view consistent. This limits the performance of prior methods that use 2D features lifted to 3D in a zero-shot fashion. However, our SoTA results (in Table 1) demonstrate that **3D-JEPA effectively learns to deal with such noise**. Qualitatively, we find that 3D-JEPA features are smoother and more consistent than 2D lifted features (as shown in Figure 1), demonstrating (again) that our method is robust to such issues.

---

> > ### Comment · Reviewer_J9xs · 2025-04-09
> >
> > Thanks for the authors' response, which has addressed my main concerns. And I will keep green light for this submission.

---

### Decision · Program_Chairs · 2025-05-01

**Decision:**

Accept (spotlight poster)

**Comment:**

The ratings for this paper are two accepts and two weak accepts. All reviewers think the paper has valuable contributions. The major concerns are about insufficient experiments and more discussions. The rebuttal has satisfactorily addressed the major concerns, and two reviewers raised their ratings. Therefore, the AC recommend acceptance.